# Reward Dimension Reduction for Scalable Multi-Objective Reinforcement Learning

**Giseung Park, Youngchul Sung** [*]
School of Electrical Engineering
Korea Advanced Institute of Science and Technology (KAIST)
Daejeon 34141, Republic of Korea
{gs.park,ycsung}@kaist.ac.kr

## Abstract

In this paper, we introduce a simple yet effective reward dimension reduction method to tackle the scalability challenges of multi-objective reinforcement learning algorithms. While most existing approaches focus on optimizing two to four objectives, their abilities to scale to environments with more objectives remain uncertain. Our method uses a dimension reduction approach to enhance learning efficiency and policy performance in multi-objective settings. While most traditional dimension reduction methods are designed for static datasets, our approach is tailored for online learning and preserves Pareto-optimality after transformation. We propose a new training and evaluation framework for reward dimension reduction in multi-objective reinforcement learning and demonstrate the superiority of our method in environments including one with sixteen objectives, significantly outperforming existing online dimension reduction methods.

## 1 Introduction

Reinforcement Learning (RL) is a powerful machine learning paradigm focused on training agents to make sequential decisions by interacting with their environment. Through trial and error, RL algorithms allow agents to iteratively improve their decision-making policies, with the ultimate goal of maximizing cumulative rewards. In recent years, the field of Multi-Objective Reinforcement Learning (MORL) has gained considerable attention due to its relevance in solving real-world control problems involving multiple, often conflicting, objectives. These problems span across domains such as advanced autonomous control (Weber et al., 2023), power system management, and logistics optimization (Hayes et al., 2022), where balancing trade-offs among competing objectives is crucial (Roijers et al., 2013).

MORL extends the traditional RL framework by enabling agents to handle multiple objectives simultaneously. This requires methods capable of identifying and managing trade-offs among these objectives. MORL specifically focuses on learning a set of policies that approximate the Pareto frontier, representing solutions where no objective can be improved without compromising others. Most of the current approaches scalarize vector rewards into scalar objectives to generate a diverse set of policies (Abels et al., 2019; Yang et al., 2019; Xu et al., 2020; Basaklar et al., 2023; Lu et al., 2023), thereby avoiding the need for retraining during the test phase.

Although these methods have proven effective in standard MORL benchmarks (Felten et al., 2023), most benchmarks involve only two to four objectives, leaving open the question of whether existing MORL algorithms can scale effectively to environments with more objectives (Hayes et al., 2022). Indeed, various practical applications demand optimizing many objectives simultaneously (Li et al., 2015). For example, Fleming et al. (2005) introduced an example of optimizing a complex jet engine control system that requires balancing eight physical objectives. In military contexts, a commander should manage dozens of objectives that directly influence decision-making (Dagistanli & Üstün, 2023), including the positions of allies and enemies, casualty rates, combat capabilities of allies and enemies, and time estimations for achieving strategic goals. When planning for multiple potential

---

[*]Youchul Sung is the corresponding author.

battle scenarios, exploring such high-dimensional objective space in its raw form is inefficient and very challenging due to the complexity of the original space (Wang & Sebag, 2013).

An advantageous feature of many real-world MORL applications is that objectives often exhibit correlations, leading to inherent conflicts or trade-offs. For example, an autonomous vehicle must balance safety and speed, where optimizing one can compromise the other. Similarly, traffic light control must manage multiple interrelated objectives to ensure smooth traffic flow (Hayes et al., 2022). These correlations suggest that reducing the dimensionality of the reward space while preserving its essential features could be a viable strategy to make current MORL algorithms scalable to many objective spaces.

Dimension reduction techniques (Roweis & Saul, 2000; Tenenbaum et al., 2000; Zass & Shashua, 2006; Lee et al., 2007; Cardot & Degras, 2018; McInnes & Healy, 2018; Bank et al., 2020), widely used in other machine learning domains, capture the most significant features of high-dimensional data while filtering out irrelevant noise. However, typical approaches operate on static datasets, whereas RL necessitates continuous data collection during online training. This introduces a unique challenge: applying dimension reduction in MORL while retaining the essential structure of the original reward space. To our knowledge, few studies have addressed this challenge within the MORL context.

In this paper, we address these challenges by introducing a simple yet effective reward dimension reduction method that scales MORL algorithms to higher-dimensional reward spaces. We propose a new training and evaluation framework tailored for the online reward dimension reduction setting. Our approach ensures that Pareto-optimality is preserved after transformation, allowing the agent to learn and execute policies that remain effective in the original reward space.

Our contributions are as follows. First, we propose a new training and evaluation framework for online reward dimension reduction in MORL. We also derive conditions and introduce learning techniques to ensure that online training and Pareto-optimality are maintained, providing a stable and efficient approach for scalable MORL. Lastly, our method demonstrates superior performance compared to existing online dimension reduction methods in MORL environments including one with sixteen objectives.

## 2 BACKGROUND

A multi-objective Markov decision process (MOMDP) is defined by the tuple $\langle \mathcal{S}, \mathcal{A}, P, \mu_0, r, \gamma \rangle$. Here, $\mathcal{S}$ represents the set of states, $\mathcal{A}$ the set of actions, $P$ the state transition probabilities, $\mu_0$ the initial state distribution, $r$ the reward function, and $\gamma \in [0, 1)$ the discount factor. Unlike the traditional single-objective MDP, the reward function $r : \mathcal{S} \times \mathcal{A} \to \mathbb{R}^K$ in a MOMDP is vector-valued, where $K \geq 2$ is the number of objectives. This vector-valued nature of the reward function allows the agent to receive multiple rewards for each state-action pair, each corresponding to a different objective.

In the context of MORL, the performance of a policy $\pi$ is evaluated by its expected cumulative reward, denoted as $J(\pi) = (J_1(\pi), \cdots, J_K(\pi)) := \mathbb{E}_\pi \left[ \sum_{t=0}^{\infty} \gamma^t r_t \right] \in \mathbb{R}^K$. To compare vector-valued rewards, we use the notion of Pareto-dominance (Roijers et al., 2013), denoted $>_P$. For two vector returns, $J(\pi)$ and $J(\pi')$, we have:

$$J(\pi') >_P J(\pi) \iff (\forall i \in \{1, \ldots, K\}, J_i(\pi') \geq J_i(\pi)) \text{ and } (\exists j \in \{1, \ldots, K\}, J_j(\pi') > J_j(\pi)).$$
$$(1)$$

This means that $J(\pi')$ Pareto-dominates $J(\pi)$ if it is at least as good as $J(\pi)$ in all objectives and strictly better in at least one.

The goal of MORL is to identify a policy $\pi$ whose $J(\pi)$ lies on the Pareto frontier (or boundary) $\mathcal{F}$ of all achievable return tuples $\mathcal{J} = \{(J_1(\pi), \cdots, J_K(\pi)) \mid \pi \in \Pi\}$, where $\Pi$ denotes the set of all possible policies. The formal definition of the Pareto frontier [1] is as follows (Roijers et al., 2013;

---

[1] Strictly speaking, the Pareto frontier can also be defined as the set of non-dominated *policies*, $\{\pi \in \Pi \mid \nexists \pi' \text{ s.t. } J(\pi') >_P J(\pi)\}$ (Hayes et al., 2022), rather than the set of non-dominated *vector returns* as shown in equation 2 . In this case, multiple policies may achieve the same vector return (Hayes et al., 2022). To avoid this redundancy, in this paper, we define the Pareto frontier and the convex coverage set as presented in equation 2 and equation 3, respectively.

Yang et al., 2019):

$$\mathcal{F} = \{J(\pi) \mid \nexists \pi' \text{ s.t. } J(\pi') >_P J(\pi)\}. \tag{2}$$

In other words, no single policy achieving $\mathcal{F}$ can improve one objective without sacrificing at least one other objective. Finding a policy achieving the Pareto frontier ensures an optimal balance among the competing objectives with the best possible trade-offs.

Researchers are also interested in obtaining policies that cover the *convex coverage set* (CCS) of a given MOMDP defined as follows (Yang et al., 2019):

$$CCS = \{J(\pi) \mid \exists \omega \in \Delta^K \text{ s.t. } \omega^\top J(\pi) \geq \omega^\top J(\pi'), \ \forall \pi' \in \Pi \text{ with } J(\pi') \in \mathcal{F}\} \tag{3}$$

where $\Delta^K$ is the $(K-1)$-simplex and $\omega \in \Delta^K$ represents a preference vector that specifies the relative importance of each objective (i.e., $\sum_{k=1}^{K} \omega_k = 1, \omega_k \geq 0, \forall k$).

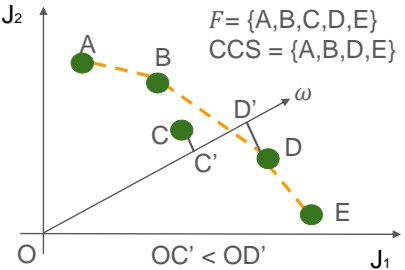

Figure 1 illustrates the relationship between Pareto frontier and CCS. In Figure 1, we assume that the achievable points $\{A, B, C, D, E\}$ form the Pareto frontier. Then, for the preference vector $\omega$ in Figure 1, the inner product between $\omega$ and the return vector at the point C is smaller than the inner product between $\omega$ and the return vector at the point D. The inner product between $\omega$ and the return vector at the point C is smaller than that between $\omega$ and any other point in the Pareto frontier. Hence, the point C is not included in the CCS. Note that the CCS represents the set of achievable returns that are optimal for some linear combination of objectives, and it is a subset of the Pareto frontier $\mathcal{F}$ by definition. Since the weighted sum is widely used in real-world applications to express specific preferences over multiple objectives (Hayes et al., 2022), CCS is a proper refinement of the Pareto frontier.

Figure 1: Comparison of the Pareto frontier $\mathcal{F}$ and the CCS for $K = 2$, where $C'$ and $D'$ represent the projections of points $C$ and $D$ onto the preference vector $\omega$, respectively. Yellow dashed line represents the outer convex boundary of $\mathcal{F}$.

In the context of *multi-policy* MORL (Roijers et al., 2013), the goal is to find multiple policies that cover (an approximation of) either the Pareto frontier or the CCS so that during test phases, we perform well across various scenarios without having to retrain from scratch. Specifically, we aim to achieve Pareto-optimal points that *maximize the hypervolume while minimizing the sparsity* (Hayes et al., 2022).

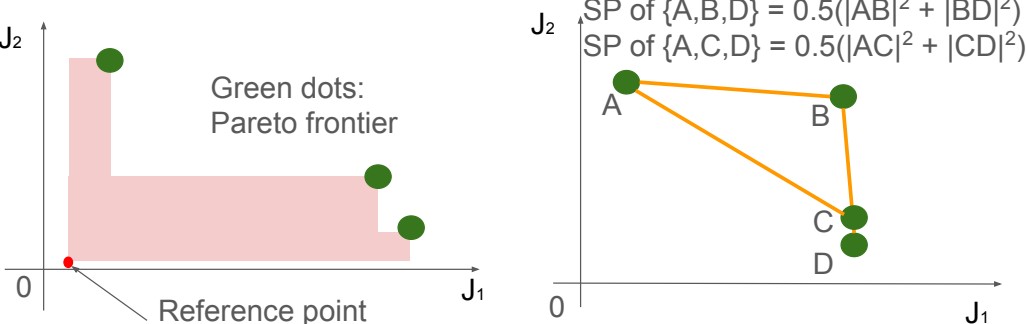

Figure 2: Evaluation metrics in multi-policy MORL: hypervolume and sparsity. (Left) Hypervolume is represented by the pink area in the figure. (Right) The sparsity of the solution set $\{A, B, D\}$ is lower than that of $\{A, C, D\}$ when points $C$ and $D$ are close, indicating that $\{A, B, D\}$ offers a more diverse set of solutions than $\{A, C, D\}$.

As seen in the left figure of Figure 2, the hypervolume measures the volume in the objective space dominated by the set of current Pareto frontier points and bounded by a reference point. In the figure, the hypervolume corresponds to the area of the pink region. This metric provides a scalar value quantifying how well the policies cover the objective space. Formally, let $X = \{x_1, \cdots, x_N\} \subset \mathbb{R}^K$ be a set of $N$ Pareto frontier points and $x_0 \in \mathbb{R}^K$ be a reference point,

where $x_i = (x_{i1}, \ldots, x_{iK})$, $i = 0, \ldots, N$. Then, the hypervolume metric $HV(X, x_0)$ is defined by the $K$-dimensional volume of the union of hybercubes $\bigcup_{i=1}^{N} \mathcal{C}_{k=1}^{K}[x_{0k}, x_{ik}]$, where $\mathcal{C}_{k=1}^{K}[x_{0k}, x_{ik}]$ is the hypercube of which side at the $k$-th dimension is given by the line segment $[x_{0k}, x_{ik}]$.

Another metric is sparsity, which assesses the distribution of policies within the objective space. As seen in the right figure of Figure 2, a set of Pareto frontier points with low sparsity ensures that the solutions are well-distributed, offering a diverse range of trade-offs among the objectives. If there is a set of $N$ Pareto frontier points $X = \{x_1, \cdots, x_N\} \subset \mathbb{R}^K$ with $x_i = (x_{i1}, \ldots, x_{iK})$ $(i = 1, \ldots, N)$, sparsity is defined as:

$$SP(X) := \frac{1}{N-1} \sum_{k=1}^{K} \sum_{i=1}^{N-1} (S_k[i] - S_k[i+1])^2 \tag{4}$$

where $S_k = \text{Sort}\{x_{ik}, 1 \leq i \leq N\}$ in descending order in the $k$-th objective, $1 \leq k \leq K$. Given a dimension $k$ and its two endpoints, $S_k[1]$ and $S_k[N]$, the Cauchy–Schwarz inequality implies that $\sum_{i=1}^{N-1} (S_k[i] - S_k[i+1])^2$ is minimized when the differences $S_k[i] - S_k[i+1]$ are constant for all $1 \leq i \leq N-1$. Therefore, sparsity acts as an indicator of how well-distributed a set of return vectors is. Reducing sparsity while maintaining a high hypervolume helps avoid situations where only a few objectives perform well. Therefore, considering both low sparsity and high hypervolume offers a more comprehensive evaluation criterion than relying solely on hypervolume.

## 3 RELATED WORK

There are mainly two branches in MORL. The first branch is single-policy MORL, where the goal is to obtain an optimal policy $\pi^* = \arg\max_\pi h(J(\pi))$ where $h : \mathbb{R}^K \to \mathbb{R}$ is a *fixed* non-decreasing utility function, mostly for non-linear one (Siddique et al., 2020; Park et al., 2024). The other branch is the multi-policy MORL, where we aim to acquire multiple policies that cover an approximation of the Pareto frontier or CCS. Beyond several classical methods such as iterative single-policy approaches (Roijers et al., 2014), current approaches in the multi-policy MORL either train a set of multiple policies (Xu et al., 2020) or train a single network to cover multiple policies (Abels et al., 2019; Yang et al., 2019; Basaklar et al., 2023; Lu et al., 2023). For completeness, these approaches should be followed by a preference elicitation method for the test phase given that additional interactive approaches are allowed (Hayes et al., 2022) (e.g., Zintgraf et al. (2018) inferred unknown user preference using queries of pairwise comparison on the Pareto frontier). Nonetheless, the area of elicitation has received far less attention than the learning methods themselves. Researchers usually focus solely on the learning algorithms during the training phase assuming that test preferences will be explicitly given.

Xu et al. (2020) trains a set of multiple policies in parallel using the concept of evolutionary learning, and the best policy in the policy set is used for evaluation during the test phase. Other works construct a single policy network parameterized by $\omega \in \Delta^K$ to cover the CCS, which is easier than direct parameterization over the set of non-decreasing functions to cover the Pareto frontier. Abels et al. (2019) and Yang et al. (2019) constructed single-policy networks to exploit the advantages of CCS. Specifically, Yang et al. (2019) defined the optimal multi-objective action-value function for all $\omega \in \Delta^K$: $Q^*(s, a, \omega) = \mathbb{E}_{P, \pi^*(\cdot|\cdot;\omega)|s_0=s, a_0=a}[\sum_{t=0}^{\infty} \gamma^t r_t] \in \mathbb{R}^K$, where the optimal policy $\pi^*$ is given by $\pi^*(\cdot|\cdot;\omega) = \arg\sup_\pi \omega^\top \mathbb{E}_{P,\pi}[\sum_{t=0}^{\infty} \gamma^t r_t]$. Based on a new definition of the multi-objective optimality operator, the authors proposed an algorithm for training a neural network $Q_\theta(s, a, \omega)$ to approximate $Q^*(s, a, \omega)$. Basaklar et al. (2023) modified the multi-objective optimality operator to match each direction of the learned action-value function and preference vector, and Lu et al. (2023) tackled a learning stability issue of multi-policy MORL by providing theoretical analysis on linear scalarization.

While these methods have demonstrated promising performance in MORL benchmarks with two to four objectives, it remains an open question whether current algorithms can effectively scale to environments with more objectives (Hayes et al., 2022). The challenge lies in effectively covering all possible preferences during training. In most previous MORL algorithms, agents sample random preferences in each episode to collect diverse behaviors. However, performing this sampling naively in high-dimensional spaces becomes computationally expensive because the coverage (or hypervolume) grows exponentially with the number of objectives (Wang & Sebag, 2013).

In this paper, we address the scalability issue by proposing a reward dimension reduction technique with a suitable training and evaluation framework to narrow down the search space while preserving the most relevant information. Our approach is motivated by the observation that objectives are correlated in many real-world cases. While a variety of dimension reduction techniques exist in machine learning (Roweis & Saul, 2000; Tenenbaum et al., 2000; Lee et al., 2007; McInnes & Healy, 2018), most are designed for static (batch-based) datasets. Only a few methods are suitable for online settings, and in some cases, no online version exists at all (McInnes & Healy, 2018). Developing online variants of batch-based dimension reduction techniques is itself an active area of research. Currently, incremental principal component analysis (PCA) and online autoencoders are commonly used for online dimension reduction (Cardot & Degras, 2018; Bank et al., 2020). However, we will demonstrate that they fail to preserve Pareto-optimality after transformation in the context of multi-policy MORL.

To our knowledge, few studies have explored reward dimension reduction in MORL. For instance, Giuliani et al. (2014) applied non-negative principal component analysis (NPCA) to a fixed set of return vectors collected from several pre-defined scenarios, identifying the principal components. However, they did not perform any further online interactions, but multi-policy MORL algorithms require online learning. In this paper, we propose a simple yet stable method for online dimension reduction that preserves Pareto-optimality after transformation, as described in the following section.

## 4 METHOD

### 4.1 TRAINING AND EVALUATION FRAMEWORK

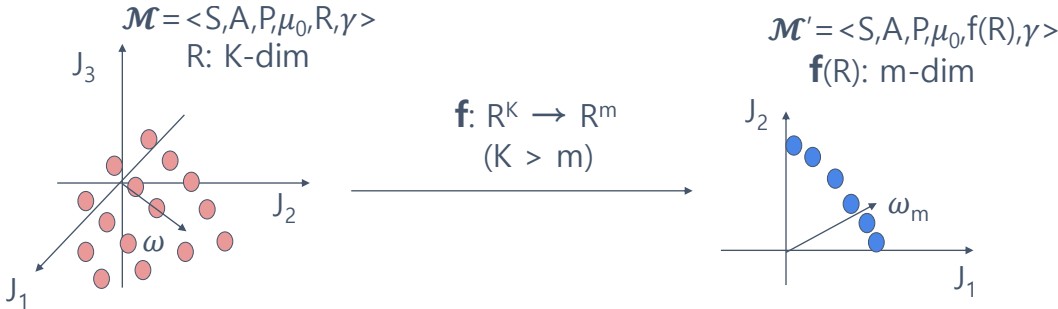

Figure 3: Our proposed reward dimension reduction framework. We design a mapping function $f : \mathbb{R}^K \to \mathbb{R}^m$ from the original reward space to the reduced reward space.

As seen in Figure 3, we aim to design a mapping function $f : \mathbb{R}^K \to \mathbb{R}^m$ for reward dimension reduction, where $K > m \geq 2$. $f$ transforms the original MOMDP $\mathcal{M} = \langle \mathcal{S}, \mathcal{A}, P, \mu_0, r, \gamma \rangle$ into another MOMDP $\mathcal{M}' = \langle \mathcal{S}, \mathcal{A}, P, \mu_0, f(r), \gamma \rangle$, reducing the dimensionality of the reward space while preserving essential features. We assume that standard multi-policy MORL approaches, such as Yang et al. (2019), perform adequately in the reduced-dimensional reward space of $\mathcal{M}'$. Then for any preference vector $\omega_m \in \Delta^m$, the $(m-1)$-simplex, the optimal multi-objective action-value function and the optimal policy are defined as:

$$Q_m^*(s, a, \omega_m) = \mathbb{E}_{P, \pi_m^*(\cdot|\cdot;\omega_m)|s_0=s, a_0=a} \left[ \sum_{t=0}^{\infty} \gamma^t f(r_t) \right] \in \mathbb{R}^m, \text{ where} \tag{5}$$

$$\pi_m^*(\cdot|\cdot;\omega_m) = \arg\sup_{\pi} \omega_m^\top \mathbb{E}_{P,\pi} \left[ \sum_{t=0}^{\infty} \gamma^t f(r_t) \right] = \arg\sup_{\pi} \mathbb{E}_{P,\pi} \left[ \sum_{t=0}^{\infty} \gamma^t (\omega_m^\top f(r_t)) \right]. \tag{6}$$

$\pi_m^*$ is also expressed as $\pi_m^*(a|s, \omega_m) = 1$ if $a = \arg\max_{a'} \omega_m^\top Q_m^*(s, a', \omega_m), \pi_m^*(a|s, \omega_m) = 0$ otherwise. Note that the key aspect of multi-policy MORL is that we learn the action-value function $Q_m^*(s, a, \omega_m)$ not just for a particular linear combination weight $\omega_m$ but for all possible weights $\{\omega_m \in \Delta^m\}$ in the training phase so that we can choose the optimal policy for any arbitrary combination weight depending on the agent's preference in the test or evaluation phase.

Our goal is to design a dimension reduction function, $f$, such that the policies learned in the reduced-dimensional space achieve high performance in the **original reward space** while satisfying two key requirements: (i) *online updates for dimension reduction* and (ii) *the preservation of Pareto-optimality* in the sense that $\forall \omega_m \in \Delta^m$,

$$\mathbb{E}_{\pi_m^*(\cdot|\cdot,\omega_m)}\left[\sum_{t=0}^{\infty}\gamma^t f(r_t)\right] \in CCS_m \Rightarrow \mathbb{E}_{\pi_m^*(\cdot|\cdot,\omega_m)}\left[\sum_{t=0}^{\infty}\gamma^t r_t\right] \in \mathcal{F} \tag{7}$$

where $CCS_m$ represents the convex coverage set in the reduced reward space and $\mathcal{F} \subset \mathbb{R}^K$ represents the Pareto frontier in the original reward space.

To the best of our knowledge, research on reward dimension reduction in MORL is limited, and there is no well-established evaluation protocol for this task. In this section, we propose a new training and evaluation framework tailored to the reward dimension reduction problem, along with the algorithm itself (outlined in Section 4.2). This framework facilitates a fair comparison of online dimension reduction techniques within the context of the original MOMDP.

During the *training phase*, we aim to learn the optimal multi-objective action-value function $Q_m^*(s, a, \omega_m)$ while we simultaneously update the dimension reduction function $f$ online. For the action-value function update, we sample data $(s, a, r, s')$ from a replay buffer but utilize the reduced-dimensional rewards $f(r)$ instead of the original rewards $r$. Our goal is to ensure that $\mathbb{E}_{\pi_m^*(\cdot|\cdot,\omega_m)}[\sum_{t=0}^{\infty}\gamma^t f(r_t)] \in CCS_m$ after the training phase ends.

In the *evaluation phase*, the learned policy $\pi_m^*(\cdot|\cdot,\omega_m)$ is tested on a set of $N_e$ preferences $\Omega_{m,N_e} \subset \Delta^m$, with $N_e = |\Omega_{m,N_e}|$, where the points are evenly distributed on the $(m-1)$-simplex. For each $\omega_m \in \Omega_{m,N_e}$, we compute the expected cumulative reward $\mathbb{E}_{\pi_m^*(\cdot|\cdot,\omega_m)}[\sum_{t=0}^{\infty}\gamma^t r_t] \in \mathbb{R}^K$ in the original reward space, as the MOMDP provides the high-dimensional vector reward at each timestep. Our goal is for the Pareto frontier points of $\{\mathbb{E}_{\pi_m^*(\cdot|\cdot,\omega_m)}[\sum_{t=0}^{\infty}\gamma^t r_t] \in \mathbb{R}^K | \omega_m \in \Omega_{m,N_e}\}$ to maximize hypervolume while minimizing sparsity.

## 4.2 Design of Dimension Reduction Function

To preserve the Pareto-optimality as shown in equation 7, we impose two minimal conditions on the dimension reduction function $f$:

**Theorem 1.** *If $f$ is affine and each element of the matrix is positive, then equation 7 is satisfied.*

*Proof.* First, if $f$ is **affine**, then $f(r) = Ar + b \in \mathbb{R}^m$, where $A \in \mathbb{R}^{m \times K}$. By linearity, $\forall \omega_m \in \Delta^m$,

$\mathbb{E}_{\pi_m^*(\cdot|\cdot,\omega_m)}[\sum_{t=0}^{\infty}\gamma^t(Ar_t+b)] = A\left(\mathbb{E}_{\pi_m^*(\cdot|\cdot,\omega_m)}[\sum_{t=0}^{\infty}\gamma^t r_t]\right) + \frac{1}{1-\gamma}b \in CCS_m.$

Next, if each element of $A$ is **positive**, we claim that $\mathbb{E}_{\pi_m^*(\cdot|\cdot,\omega_m)}[\sum_{t=0}^{\infty}\gamma^t r_t] \in \mathcal{F}$ so that the Pareto-optimality in equation 7 is preserved.

This is proved by contradiction. Given $\omega_m \in \Delta^m$, suppose $\exists \pi' \in \Pi$ s.t. $\mathbb{E}_{\pi'}[\sum_{t=0}^{\infty}\gamma^t r_t] >_P \mathbb{E}_{\pi_m^*(\cdot|\cdot,\omega_m)}[\sum_{t=0}^{\infty}\gamma^t r_t]$ in the original reward space. By the definition of $>_P$ in equation 1, each dimension of $\mathbb{E}_{\pi'}[\sum_{t=0}^{\infty}\gamma^t r_t] - \mathbb{E}_{\pi_m^*(\cdot|\cdot,\omega_m)}[\sum_{t=0}^{\infty}\gamma^t r_t] \in \mathbb{R}^K$ is non-negative and at least one dimension is positive.

For each $1 \leq j \leq m$, let $a_j^\top \in \mathbb{R}^{1 \times K}$ be the $j$-th row vector of $A$ and $b_j$ be the $j$-th element of $b$. Then $a_j^\top(\mathbb{E}_{\pi'}[\sum_{t=0}^{\infty}\gamma^t r_t] - \mathbb{E}_{\pi_m^*(\cdot|\cdot,\omega_m)}[\sum_{t=0}^{\infty}\gamma^t r_t]) > 0$. In other words, we have $A\mathbb{E}_{\pi'}[\sum_{t=0}^{\infty}\gamma^t r_t] >_P A\mathbb{E}_{\pi_m^*(\cdot|\cdot,\omega_m)}[\sum_{t=0}^{\infty}\gamma^t r_t]$ in the reduced-dimensional space. By linearity, adding $\mathbb{E}_{\pi'}[\sum_{t=0}^{\infty}\gamma^t b] = \mathbb{E}_{\pi_m^*(\cdot|\cdot,\omega_m)}[\sum_{t=0}^{\infty}\gamma^t b] = \frac{1}{1-\gamma}b$ to both sides gives

$$\mathbb{E}_{\pi'}\left[\sum_{t=0}^{\infty}\gamma^t(Ar_t+b)\right] >_P \mathbb{E}_{\pi_m^*(\cdot|\cdot,\omega_m)}\left[\sum_{t=0}^{\infty}\gamma^t(Ar_t+b)\right]. \tag{8}$$

Since $CCS_m$ is by definition a subset of the Pareto frontier in the reduced-dimensional space, $CCS_m$ consists of vector returns in the Pareto frontier. Therefore, equation 8 gives a contradiction since $\mathbb{E}_{\pi_m^*(\cdot|\cdot,\omega_m)}[\sum_{t=0}^{\infty}\gamma^t(Ar_t+b)] \in CCS_m$ is Pareto dominated by $\mathbb{E}_{\pi'}[\sum_{t=0}^{\infty}\gamma^t(Ar_t+b)]$. □

In short, the condition of $f(r) = Ar + b$ with $A \in \mathbb{R}_+^{m \times K}$ guarantees the preservation of Pareto-optimality in equation 7. From equation 6,

$$\pi_m^*(\cdot|\cdot, \omega_m) = \sup_\pi \mathbb{E}\left[\sum_{t=0}^\infty \gamma^t(\omega_m^\top A r_t)\right] + \frac{1}{1-\gamma}\omega_m^\top b = \sup_\pi \mathbb{E}\left[\sum_{t=0}^\infty \gamma^t(\omega_m^\top A r_t)\right]. \tag{9}$$

In discounted reward settings, the bias term $b$ does not affect the determination of $\pi_m^*$, so we set $b = 0$ for simplicity.

In addition, we impose another condition that $A$ is **row-stochastic**: $\sum_{k=1}^K A_{jk} = 1, 1 \le j \le m$. Then

$$\sum_{k=1}^K (A^\top \omega_m)_k = \sum_{k=1}^K \sum_{j=1}^m A_{jk}(\omega_m)_j = \sum_{j=1}^m (\omega_m)_j \sum_{k=1}^K A_{jk} = \sum_{j=1}^m (\omega_m)_j = 1. \tag{10}$$

In other words, $\forall \omega_m \in \Delta^m$, we have the corresponding preference vector $A^\top \omega_m \in \Delta^K$ in the original reward space. Let $A^\top = [a_1, \cdots, a_m] \in \mathbb{R}_+^{Km}$ where $a_j \in \Delta^K, 1 \le j \le m$. Then $A^\top \omega_m \in \Delta^K$ and the set $\{A^\top \omega_m | \omega_m \in \Delta^m\} \subset \Delta^K$ is the convex combination of $a_j \in \Delta^K, 1 \le j \le m$. Conceptually, the role of the matrix $A$ is to narrow down the preference search space from $\Delta^K$ to a proper subset of $\Delta^K$.

The next question is "How should we design the affine transform $A$ to preserve the information of the original vector reward function $r$?" To address this question, we propose constructing a reconstruction neural network $g_\phi$, where the input is the reduced-dimensional reward $f(r)$. The network $g_\phi$ is trained to minimize the reconstruction loss:

$$\min_{A>0, \ A \text{ row-stochastic}, \phi} \mathbb{E}_{s,a}\|r(s,a) - g_\phi(f(r(s,a)))\|^2 \tag{11}$$

where $A > 0$ denotes that each element of $A$ is positive. This approach, combining compression with reconstruction, is widely employed to capture the essential features of input data while discarding irrelevant information (Baldi, 2012; Kingma & Welling, 2014; Berahmand et al., 2024). However, solving the optimization problem in equation 11 is more challenging than conventional autoencoder-style learning, where the encoder is a general neural network trained without constraints. In contrast, our method must ensure that the matrix $A$ satisfies both the positivity constraint $A > 0$ and row-stochasticity during online training.

To overcome this challenge, we introduce a novel approach by parameterizing $A$ using softmax parameterization, ensuring both positivity and row-stochasticity constraints are satisfied throughout training. Our implementation in PyTorch (Paszke et al., 2019) effectively applies this parameterization, and we solve the optimization in equation 11 using stochastic gradient descent in an online setting. The reconstruction loss is minimized alongside the training of the parameterized multi-objective action-value function $Q_\theta(s, a, \omega_m)$, which approximates $Q_m^*(s, a, \omega_m)$ as defined in equation 5. Additionally, we found that applying dropout (Srivastava et al., 2014) to $g_\phi$ during training in equation 11 further enhances overall performance.

## 5 EXPERIMENTS

### 5.1 ENVIRONMENT AND BASELINES

While various practical applications require addressing many objectives (Fleming et al., 2005; Li et al., 2015; Hayes et al., 2022), there currently exist few MORL simulation environments with reward dimensions exceeding four (Hayes et al., 2022; Felten et al., 2023). To address this issue, we considered the following two MORL environments: LunarLander-5D (Hung et al., 2023) and our modified implementation of an existing traffic light control environment (Alegre, 2019) to create a sixteen-dimensional reward setting.

LunarLander-5D is a challenging MORL environment with a five-dimensional reward function where the agent aims to land a lunar module on the moon's surface successfully. Each reward dimension represents: (i) a sparse binary indicator for successful landing (+ for success, - for crash), (ii) a combined measure of the module's position, velocity, and orientation, (iii) the fuel cost of the main engine, (iv) the fuel cost of the side engines, and (v) a time penalty. This environment

presents significant challenges because failing to balance these objectives effectively can easily lead to unsuccessful landings (Felten et al., 2023; Hung et al., 2023).

Traffic light control is a practical example of a problem that can be formulated as MORL, where efficiently balancing many correlated objectives is crucial (Hayes et al., 2022). As shown in Figure 4, the traffic intersection features four road directions (North, South, East, and West), each with four inbound and four outbound lanes. At each time step, the agent receives a state representing traffic flow information. The traffic light controller selects a phase as its action, and the reward is a sixteen-dimensional vector where each dimension corresponds to a measure proportional to the negative total waiting time of cars on the respective inbound lanes.

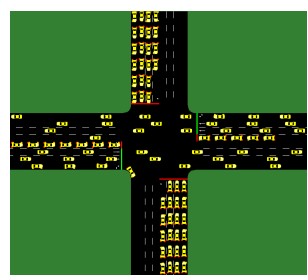

Figure 4: A snapshot of our considered environment: traffic light control.

In our experiments, we used the MORL algorithm from Yang et al. (2019) as the base algorithm for the original reward space (Base). We incorporated several online dimension reduction methods to the vector rewards in the base algorithm, including online autoencoder (AE) (Bank et al., 2020), incremental PCA (Cardot & Degras, 2018), our online implementation of conventional batch-based NPCA (Zass & Shashua, 2006), and our proposed approach. We followed the training and evaluation framework outlined in Section 4.1, setting $m = 16$ when evaluating the base algorithm alone.

For incremental PCA, we update the sample mean vector $\mu \in \mathbb{R}^K$ and the sample covariance matrix $C \in \mathbb{R}^{K \times K}$ at each timestep $t$ using vector reward $r_t$. We periodically perform eigendecomposition on $C$ and select the top $m$ eigenvectors $u_1, \ldots, u_m \in \mathbb{R}^K$ corresponding to the largest eigenvalues, maximizing $\sum_{l=1}^{m} u_l^\top C u_l$. We construct the matrix $U = [u_1, \ldots, u_m] \in \mathbb{R}^{K \times m}$ so that $U^\top(r - \mu) \in \mathbb{R}^m$ represents the reduced vector for $r$, following the PCA assumption that the transformed vectors are centered (Cardot & Degras, 2018).

For the online implementation of original NPCA (Zass & Shashua, 2006), we directly parameterize $U = [u_1, \ldots, u_m] \in \mathbb{R}^{K \times m}$ with a non-negativity constraint for efficient training (which we denote as $U \geq 0$). This direct parameterization removes an extra hyperparameter tuning for the constraint $U \geq 0$ and gives a fair implementation compared with our method that also uses direct parameterization. We optimize the objective $\max_{U \geq 0} \sum_{l=1}^{m} u_l^\top C u_l - \beta \|U^\top U - I_m\|^2$ using gradient descent, with a hyperparameter $\beta > 0$. Balancing the PCA loss with the orthonormality constraint creates a trade-off between capturing principal component information and maintaining orthonormal basis vectors. Both PCA and NPCA do not use reconstructor $g_\phi$.

In the traffic environment, we set $m = 4$ for all online dimension reduction methods. We set $m = 3$ for LunarLander-5D. To enhance the statistical reliability of our experimental results, we applied a 12.5% trimmed mean by excluding the seeds with maximum and minimum hypervolume values over eight random seeds and reporting the averages of the metrics over the remaining six random seeds. This offers better robustness against outliers than the standard average (Maronna et al., 2019). (Additional implementation details can be found in Appendix E.)

## 5.2 RESULTS

| | Base | PCA | AE | NPCA | **Ours** |
|---|---|---|---|---|---|
| **HV**($\times 10^7, \uparrow$) | $3.1 \pm 4.7$ | $3.2 \pm 4.2$ | $0$ | $1.7 \pm 3.1$ | **$25.6 \pm 6.9$** |
| **SP**($\times 10^2, \downarrow$) | $31.2 \pm 25.3$ | $188.6 \pm 180.7$ | $31.3 \pm 30.6$ | $53.0 \pm 47.7$ | **$1.1 \pm 1.2$** |

Table 1: Performance comparison in LunarLander-5D environment, with the reference point for hypervolume evaluation set to $(0, -100, -100, -100, -100) \in \mathbb{R}^5$. HV: hypervolume, SP: sparsity.

Table 1 demonstrates that in the LunarLander-5D environment, our algorithm outperforms the baseline methods in both hypervolume and sparsity metrics. Specifically, our approach improves the base algorithm's hypervolume by a factor of 8.3. It also reduces sparsity to the ratio of $\frac{1}{28.4}$, resulting in

more diverse and better-performing solutions. Note that the hypervolume values reflect successful landing episodes, so our dimension reduction is more efficient for achieving successful landing and balancing remaining objectives simultaneously than the baselines.

| | Base | PCA | AE | NPCA | **Ours** |
|---|---|---|---|---|---|
| **HV**($\times 10^{61}, \uparrow$) | $4.4 \pm 6.8$ | $0$ | $0.007 \pm 0.018$ | $19.4 \pm 15.3$ | $\mathbf{166.9 \pm 48.1}$ |
| **SP**($\times 10^{5}, \downarrow$) | $1842 \pm 1290$ | $3837 \pm 2164$ | $7834 \pm 3323$ | $34.2 \pm 52.3$ | $\mathbf{2.3 \pm 1.0}$ |

Table 2: Performance comparison in our traffic experiment where we set reference point for hypervolume evaluation to $(-10^4, -10^4, \cdots, -10^4) \in \mathbb{R}^{16}$. HV: hypervolume, SP: sparsity.

Next, Table 2 demonstrates that our algorithm consistently outperforms the baseline methods in the traffic environment with sixteen-dimensional reward. Our algorithm improves the base algorithm's hypervolume by a factor of 37.9 while significantly reducing sparsity, indicating that reward dimension reduction effectively scales the base algorithm to higher-dimensional spaces. The PCA-based dimension reduction is an affine transformation, but because the matrix does not meet the positivity condition in Theorem 1, it fails to guarantee Pareto-optimality as outlined in equation 7. Similarly, the AE method uses a nonlinear transformation that fails to satisfy the linearity requirement in Theorem 1, producing worse hypervolume and sparsity than the base case.

| | NPCA | NPCA-ortho | **Ours** |
|---|---|---|---|
| **HV**($\times 10^{61}, \uparrow$) | $19.4 \pm 15.3$ | $0.3 \pm 0.5$ | $\mathbf{166.9 \pm 48.1}$ |
| **SP**($\times 10^{5}, \downarrow$) | $34.2 \pm 52.3$ | $203.7 \pm 24.1$ | $\mathbf{2.3 \pm 1.0}$ |
| **Rank** | 1 | 4 | 4 |

Table 3: Performance comparison in the traffic experiment with NPCA and NPCA-ortho where "Rank" refers to the rank of the matrix in each method.

In Table 2, our method outperforms NPCA by significantly increasing hypervolume and reducing sparsity, with improvements of 8.6x in hypervolume. Although NPCA employs an affine transformation with a nonnegative matrix, its online variant encounters instability due to the conflicting objectives of optimizing the principal component loss while maintaining the orthonormality constraint. As shown in Table 3, the best-performing NPCA models, in terms of hypervolume and sparsity, had matrices of rank 1. The learning process prioritized maximizing the PCA loss, at the expense of enforcing the orthonormality constraint, producing completely overlapping basis vectors.

To address this, we tuned hyperparameters to emphasize the orthonormality constraint, denoting this variant as NPCA-ortho. However, Table 3 shows that this adjustment led to a performance decline compared to NPCA. The reason is that assigning more weight to the orthonormality constraint weakened the PCA update, significantly reducing its ability to capture relevant information from the original reward space. Additionally, we found that balancing the two losses was highly sensitive and difficult to fine-tune. In contrast, our method avoids these issues, offering a more stable and effective solution for reward dimension reduction without the trade-offs inherent in NPCA's design.

To better illustrate results in high-dimensional space, we visualized the Pareto frontier points obtained from the traffic environment using t-SNE (Van der Maaten & Hinton, 2008), as detailed in Appendix B.1. We also present hypervolume values for different reference points and an additional metric called Expected Utility Metric (EUM) (Zintgraf et al., 2015; Hayes et al., 2022) in Appendix B.2 and B.3, respectively.

## 5.3 ABLATION STUDY

In this section, we investigate the impact of key components in our proposed dimension reduction approach. First, we analyze the effect of the constraints imposed on the dimension reduction function $f$. Specifically, we examine three aspects: bias, row-stochasticity, and positivity to assess their influence on preserving Pareto-optimality. Next, we evaluate the effect of applying dropout to the reconstructor $g_\phi$ in equation 11. We conducted our ablation study in the traffic environment.

Table 4 presents the results of our first ablation study. Adding a bias term to $f$ results in a slight decrease in hypervolume and an increase in sparsity compared to our method. However, the impact is

|  | **Ours** | +bias | -rowst | -positivity | -rowst, -positivity |
|---|---|---|---|---|---|
| **HV**($\times 10^{61}, \uparrow$) | **166.9** | 132.9 | 46.8 | 0 | 0 |
| **SP**($\times 10^5, \downarrow$) | **2.3** | 2.7 | 38.8 | 4066.6 | 5310.7 |

Table 4: Ablation study examining the effect of different conditions on the dimension reduction function $f$. "+bias" adds a bias term $b$ in $f$; "-rowst" removes the row-stochasticity constraint while retaining the positivity condition; "-positivity" removes the positivity condition.

less severe than the other modifications. While, in theory, the bias term $b$ does not affect the determination of the optimal policy under discounted reward settings (as shown in equation 9), in practice, introducing a bias term offers minimal benefit, so a purely linear transformation is sufficient.

We next observe a performance drop when the row-stochasticity condition is removed. Notably, sparsity increased sharply by a factor of 16.9, highlighting the detrimental impact of this removal. Note that the direction of each preference vector $\omega_m$, not the magnitude, matters for the determination of optimal policy $\pi_m^*$ in equation 6. By confining the search space to the simplex, the learning process can focus on finding the correct direction to extract essential reward information, rather than expending unnecessary effort on adjusting magnitudes. Consequently, enforcing the row-stochasticity constraint enhances learning efficiency, leading to more diverse solutions.

If we remove the positivity condition while maintaining the row-stochasticity constraint, the algorithm produces zero hypervolume. This is due to the lack of the positivity condition required by Theorem 1. Finally, further removing the row-stochasticity gives $f(r) = Ar$ with a generic linear matrix $A$ that also fails to preserve Pareto-optimality in equation 7.

In summary, the positivity condition is essential for maintaining Pareto-optimality, while the row-stochasticity constraint improves the efficiency of online learning under the positivity condition.

|  | **Ours** | Without dropout |
|---|---|---|
| **HV**($\times 10^{61}, \uparrow$) | **166.9** | 127.1 |
| **SP**($\times 10^5, \downarrow$) | **2.3** | 9.5 |

Table 5: Ablation study on the effect of applying dropout to the reconstructor $g_\phi$ in equation 11.

Next, we evaluated the effect of applying dropout to $g_\phi$ in equation 11. As shown in Table 5, omitting dropout resulted in a slight decrease in hypervolume, while sparsity increased by a factor of 4.1. Dropout can be interpreted as a form of regularization, approximately equivalent to $L_2$ regularization after normalizing the input vectors (Wager et al., 2013). $L_2$ regularization helps prevent certain weights in a neural network from becoming excessively large.

In our context, each output feature of the reconstructor $g_\phi$ corresponds to an objective in MORL, and there are implicit correlations among these $K$ objectives. By applying dropout, we mitigate the risk of the model prematurely focusing on a subset of the $K$ objectives by preventing the weights of $g_\phi$ from growing excessively large. This explains why sparsity increases when dropout is not used: the model may lose the opportunity to generate diverse solutions along the Pareto frontier, as learning tends to focus prematurely on a subset of the $K$ objectives without weight regularization. (We also provide an ablation study on the effect of the reduced dimensionality $m$ in Appendix C.)

## 6 CONCLUSION AND FUTURE WORK

In this paper, we proposed a simple yet effective reward dimension reduction technique to address the scalability challenges of multi-policy MORL algorithms. Our dimension reduction method efficiently captures the key features of the reward space, enhancing both learning efficiency and policy performance while preserving Pareto-optimality during online learning. We also introduced a new training and evaluation framework tailored to reward dimension reduction, demonstrating superior performance compared to existing methods. Our future work includes developing more benchmarks and informative metrics for high-dimensional MORL scenarios to gain deeper insight and further advance the field.

## REPRODUCIBILITY STATEMENT

We provide detailed descriptions of each algorithm in Section 5.1 and Appendix E, including the techniques, fine-tuned hyperparameters, and infrastructures used in our experiments. The evaluation protocol for performance comparison is described in Section 4.1. Furthermore, Theorem 1 is self-contained, so it is easy to verify the theoretical results. The link to our code is `https://github.com/Giseung-Park/Dimension-Reduction-MORL`.

## ACKNOWLEDGMENTS

This work was supported in part by Institute of Information & Communications Technology Planning & Evaluation (IITP) grant funded by the Korea government (MSIT) (No. RS-2024-00457882, AI Research Hub Project) and in part by Institute of Information & Communications Technology Planning & Evaluation (IITP) grant funded by the Korea government (MSIT) (No. 2022-0-00469, Development of Core Technologies for Task-oriented Reinforcement Learning for Commercialization of Autonomous Drones).

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

## A  LIMITATION AND FUTURE WORK

While our approach offers a promising solution for scaling MORL algorithms, several avenues remain for future research. First, as discussed in Section 5.1, the lack of benchmarks for environments with more than ten objectives limits the comprehensive validation of our method. Developing robust benchmarks for high-dimensional MORL scenarios is a crucial direction for our future research.

Second, recent research in MORL has focused on developing additional metrics to better evaluate performance, recognizing that the data behavior in MORL is more complex than in standard RL. High-dimensional scenarios are difficult to visualize, and data behavior often deviates from intuitive expectations (Lee et al., 2007). Therefore, designing informative metrics beyond standard measures like hypervolume and sparsity is essential for gaining deeper insights and advancing the field.

Third, although we provided mathematical conditions for preserving Pareto-optimality in Theorem 1, these are only sufficient conditions. We investigated the effect of each condition in Section 5.3. However, our method's theoretical guarantees will be more solid if we establish the necessary conditions that pinpoint when Pareto-optimality fails. We provide a detailed discussion in Appendix D.

Fourth, while our approach enables effective training for scalable MORL, for practical use, test preference vectors in their original high-dimensional form must be reduced to the lower-dimensional space learned by our model. Developing methods for preference vector reduction, and potentially integrating preference elicitation mentioned in Section 3, will be essential for making our approach more practical and complete.

Lastly, our method can be extended in various directions. For example, constrained MORL represents a promising direction, especially for safety-critical tasks where additional constraints must be considered. This extension could open up new applications where optimality and safety are paramount. Also, combining reward dimension reduction with reward canonicalization (Gleave et al., 2020) and extending reward linear shifting (Sun et al., 2022) to high-dimensional offline MORL represent promising avenues for extending our work, both theoretically and experimentally.

# B    ADDITIONAL EXPERIMENTS

## B.1    VISUALIZATION OF THE PARETO FRONTIERS

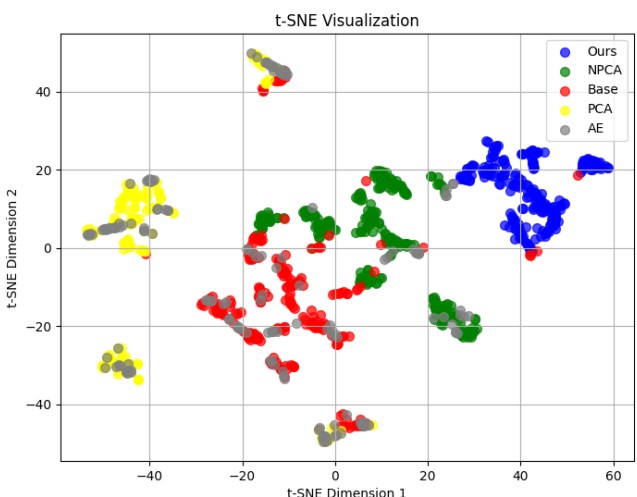

Figure 5:    t-SNE visualization of the acquired Pareto frontier points

In Figure 5, we visualized the Pareto frontier obtained in the traffic environment for each algorithm using t-SNE (Van der Maaten & Hinton, 2008). We emphasize that **our primary objective is to cover a broad region of the Pareto frontier, not merely to cover a wide region of a high-dimensional reward space itself**. Although AE solutions may appear widely distributed, this does not necessarily imply extensive coverage of the Pareto frontier because the Pareto frontier is a subset of the original space. Given that AE yields a low hypervolume, it is less likely to represent a wide range of the Pareto frontier.

According to the overlap analysis, smaller overlaps with the Base method suggest a different local structure, for example, either in a way of our method (with high hypervolume) or a way of PCA (with very low hypervolume). This qualitative difference suggests that our method and PCA are distinct in their approach to exploring the solution space. Also, NPCA overlaps with more points from the Base and AE methods than ours, demonstrating the insufficiency of NPCA in covering a larger region of the Pareto frontier than the Base method.

## B.2    HYPERVOLUMES WITH DIFFERENT REFERENCE POINTS

In our traffic environment, we selected the reference point $(-10^4, -10^4, \ldots, -10^4) \in \mathbb{R}^{16}$ based on observations that, after the initial exploration phase, most points in the current Pareto fronts fell within this defined region (except for PCA). However, some points may deviate from this region. To account for these outliers, the reference points can be adjusted accordingly. We evaluated the hypervolume using different reference points in both the traffic environment and LunarLander-5D. As shown in Tables 6 and 7, our algorithm consistently outperforms the baseline methods.

| | Base | PCA | AE | NPCA | **Ours** |
|---|---|---|---|---|---|
| **HV1**($\times 10^7, \uparrow$) | $3.1 \pm 4.7$ | $3.2 \pm 4.2$ | $0$ | $1.7 \pm 3.1$ | $\mathbf{25.6 \pm 6.9}$ |
| **HV2**($\times 10^8, \uparrow$) | $7.6 \pm 9.0$ | $8.8 \pm 7.9$ | $0$ | $4.9 \pm 6.8$ | $\mathbf{37.1 \pm 6.7}$ |

Table 6:    Performance comparison in the LunarLander-5D environment. The reference points for hypervolume evaluation are set to $(0, -100, -100, -100, -100) \in \mathbb{R}^5$ for HV1 and $(0, -150, -150, -150, -150) \in \mathbb{R}^5$ for HV2. HV: hypervolume.

| | Base | PCA | AE | NPCA | **Ours** |
|---|---|---|---|---|---|
| **HV1**$(\times 10^{61}, \uparrow)$ | $4.4 \pm 6.8$ | 0 | $0.007 \pm 0.018$ | $19.4 \pm 15.3$ | $\mathbf{166.9 \pm 48.1}$ |
| **HV2**$(\times 10^{67}, \uparrow)$ | $5.0 \pm 6.7$ | 0 | $0.3 \pm 0.7$ | $11.5 \pm 4.8$ | $\mathbf{29.3 \pm 3.4}$ |
| **HV3**$(\times 10^{73}, \uparrow)$ | $1.6 \pm 0.8$ | 0 | $0.2 \pm 0.3$ | $1.9 \pm 0.4$ | $\mathbf{2.9 \pm 0.2}$ |

Table 7: Performance comparison in the traffic experiment. The reference points for hypervolume evaluation are set to $(-10^4, -10^4, \cdots, -10^4) \in \mathbb{R}^{16}$ for HV1, $(-2 \times 10^4, -2 \times 10^4, \cdots, -2 \times 10^4) \in \mathbb{R}^{16}$ for HV2, and $(-4 \times 10^4, -4 \times 10^4, \cdots, -4 \times 10^4) \in \mathbb{R}^{16}$ for HV3. HV: hypervolume.

### B.3 DISCUSSION ON EUM METRIC

To ensure that the current Pareto frontier $\mathcal{F}$ captures a diverse set of return vectors in the original reward space, we introduce an additional metric: the Expected Utility Metric (EUM) (Zintgraf et al., 2015; Hayes et al., 2022). The EUM is defined as follows:

$$EUM(\mathcal{F}, f_s, \Omega_{K,\bar{N}_e}) := \mathbb{E}_{\omega \in \Omega_{K,\bar{N}_e}}[\max_{r \in \mathcal{F}} f_s(\omega, r)]. \tag{12}$$

Here, $\Omega_{K,\bar{N}_e} \subset \Delta^K$ represents a set of $\bar{N}_e$ preferences in the original reward space and $f_s$ is a scalarization function. We consider EUM because it effectively evaluates an agent's performance across a broad range of preferences (Hayes et al., 2022), aiming for a higher EUM to prevent the Pareto frontier from covering only a narrow region within the original reward space. To compute EUM, we define $\Omega_{K,\bar{N}_e}$ as the set of $\bar{N}_e$ normalized points evenly distributed on the $(K-1)$-simplex $\Delta^K$. We also set the scalarization function by $f_s(\omega, r) = \|\mathrm{proj}_\omega[r]\|$, representing the projected length of the vector $r$ onto $\omega \in \Omega_{K,\bar{N}_e}$. We set $\bar{N}_e$ to 126 and 15,504 for LunarLander and the traffic environment, respectively.

| LunarLander-5D | Base | PCA | AE | NPCA | **Ours** |
|---|---|---|---|---|---|
| **EUM**$(\uparrow)$ | $-25.8 \pm 24.3$ | $-20.2 \pm 21.5$ | $-76.2 \pm 48.6$ | $-28.4 \pm 13.9$ | $\mathbf{-11.5 \pm 5.4}$ |

Table 8: Performance comparison in LunarLander-5D experiment. EUM: expected utility metric.

| Traffic | Base | PCA | AE | NPCA | **Ours** |
|---|---|---|---|---|---|
| **EUM**$(\times 10^3, \uparrow)$ | $-3.4 \pm 2.9$ | $-35.1 \pm 15.2$ | $-16.1 \pm 8.8$ | $-4.4 \pm 1.2$ | $\mathbf{-2.0 \pm 1.0}$ |

Table 9: Performance comparison in our traffic experiment. EUM: expected utility metric.

As demonstrated in Tables 8 and 9, our method outperforms the baseline approaches in terms of the EUM. It is important to note that during training, the Base method uses equidistant points in the original reward space, which naturally leads to high EUM values, especially when the reward space has a high dimensionality. However, our algorithm demonstrates superior performance compared to other reward dimension reduction methods, particularly in higher-dimensional environments like the traffic scenario.

## C    EFFECT OF THE REDUCED DIMENSIONALITY

| $m$ | 2 | 4 | 6 | 8 | 10 |
|---|---|---|---|---|---|
| **HV**$(\times 10^{63}, \uparrow)$ | 1.4 | 1.7 | 1.7 | 1.9 | 1.1 |
| **SP**$(\times 10^5, \downarrow)$ | 1.5 | 2.3 | 30 | 20 | 81 |

Table 10: Ablation study on the effect of the reduced dimensionality $m$ of $f$.

Table 10 presents the effect of varying the reduced dimensionality $m$ in the traffic environment. As $m$ increases from 4 to 6, the sparsity increases significantly while the hypervolume remains unchanged, resulting in scenarios where only a few objectives perform well. When $m$ increases further from 8 to 10, both sparsity and hypervolume decrease, leading to a lower-quality set of returns in the original reward space.

$m$ is a hyperparameter for our algorithm, and selecting an appropriate value in practice is achievable. This is because we can estimate the effective rank of the sample covariance matrix recursively (Cardot & Degras, 2018) during the early exploration phase of the RL algorithm, rather than throughout the entire training process. We found that this straightforward approach performs effectively in our experiments.

## D   DISCUSSION ON THE NECESSARY CONDITION OF THEOREM 1

We acknowledge that theoretically analyzing the opposite direction of Theorem 1 is challenging. To find conditions for a counterexample of $f$ beyond $f(r) = Ar + b$ with $A \in \mathbb{R}_+^{m \times K}$, we may follow a similar flow in the proof of the Theorem 1. Given $\omega_m \in \Delta^m$, suppose $\exists \pi' \in \Pi$ s.t. $\mathbb{E}_{\pi'} \left[ \sum_{t=0}^{\infty} \gamma^t r_t \right] >_P \mathbb{E}_{\pi_m^*(\cdot|\cdot,\omega_m)} \left[ \sum_{t=0}^{\infty} \gamma^t r_t \right]$ in the original reward space. We first impose that $f = [f_1, \cdots, f_m]$ be a *monotonically strictly increasing function* satisfying $A >_P B \Rightarrow f_j(A) > f_j(B)$ for all $1 \le j \le m$ (Hayes et al., 2022). Then we have $f(\mathbb{E}_{\pi'} \left[ \sum_{t=0}^{\infty} \gamma^t r_t \right]) >_P f(\mathbb{E}_{\pi_m^*(\cdot|\cdot,\omega_m)} \left[ \sum_{t=0}^{\infty} \gamma^t r_t \right])$ in the reduced-dimensional space. If $f$ further satisfies

$$\mathbb{E}_{\pi'} \left[ \sum_{t=0}^{\infty} \gamma^t f(r_t) \right] >_P \mathbb{E}_{\pi_m^*(\cdot|\cdot,\omega_m)} \left[ \sum_{t=0}^{\infty} \gamma^t f(r_t) \right], \tag{13}$$

this gives a contradiction and $f$ becomes our target counterexample. This is directly satisfied when $f$ is affine. However, it is difficult to find such an example other than affine functions, primarily due to the inequality in $>_P$.

Alternatively, we conducted an empirical analysis by relaxing the condition in equation 13 and directly optimizing $f$ under the sole constraint of a strictly monotonically increasing function. As part of our ablation study, we parameterized $f$ as a strictly monotonically increasing function using a neural network (similar to approaches like Rashid et al. (2018) but maintaining strict monotonicity) and trained it within the traffic environment, denoted by "Monotone."

|  | **Ours** | Monotone |
|---|---|---|
| **HV**($\times 10^{61}, \uparrow$) | **166.9** | 0 |
| **SP**($\times 10^5, \downarrow$) | **2.3** | 5353.0 |

Table 11:  Ablation study on the effect of imposing strict monotonicity on $f$.

Table 11 shows that this approach resulted in a hypervolume of zero, similar to the "-positivity" and "-rowst, -positivity" cases in Table 4. This suggests that merely imposing a strictly monotonically increasing function condition is insufficient to construct a meaningful counterexample in practice. Importantly, nonzero hypervolume was only achieved when both the affine and positivity conditions were satisfied, as demonstrated in the "-rowst" case from Table 4. These results underscore the empirical effectiveness of our algorithm based on Theorem 1.

## E   IMPLEMENTATION DETAILS

### E.1   SOURCE CODE AND ENVIRONMENT

For our implementation, we adapted morl-baselines (Felten et al., 2023) and integrated it with sumo-rl (Alegre, 2019), a toolkit designed for traffic light control simulations, as discussed in Section 5. For LunarLander-5D, we used morl-baselines (Felten et al., 2023) with the reward function provided by the source code of Hung et al. (2023).

The traffic light system offers four distinct phases: (i) Straight and right turns for North-South traffic, (ii) Left turns for North-South traffic, (iii) Straight and right turns for East-West traffic, and (iv) Left turns for East-West traffic. At each time step, the agent receives a 37-dimensional state, which includes a one-hot encoded vector representing the current traffic light phase, the number of vehicles

in each incoming lane, and the number of vehicles traveling at less than 0.1 meters per second for each lane. The simulation starts with a one-hot vector where the first element is set to one. Based on this state, the controller chooses the next traffic light phase. The time between phase changes is 20 seconds, with each episode spanning 4000 seconds, or 200 timesteps. When transitioning to a different phase, the last 2 seconds of the interval display a yellow light to minimize vehicle collisions. The reward, represented by a sixteen-dimensional vector, is calculated as the negative total waiting time for vehicles on each inbound lane. The simulation runs for 52,000 timesteps in total. For LunarLander-5D, the simulation runs for 2M timesteps.

## E.2 BASELINES

For our proposed method and the baselines, we set the discount factor $\gamma = 0.99$ and use a buffer size of 52,000 and 1M for traffic and LunarLander, respectively. In Base algorithm (Yang et al., 2019), we utilize a multi-objective action-value network $Q_\theta$ with an input size of observation dimension plus $K$, two hidden layers of 128(LunarLander)/256(traffic) units each, and ReLU activations after each hidden layer. The output layer has a size of $|\mathcal{A}| \times K$. For the dimension reduction methods, the $Q_\theta$ network has an input size of input size of observation dimension plus $m$, two hidden layers of 128(LunarLander)/256(traffic) units with ReLU activations, and an output layer of size $|\mathcal{A}| \times m$.

We train $Q_\theta$ using the Adam optimizer (Kingma & Ba, 2015), applying the loss function after the first 200 timesteps, with a learning rate of 0.0003 and a minibatch size of 32. Exploration follows an $\epsilon$-greedy strategy, with $\epsilon$ linearly decaying from 1.0 to 0.05 over the first 10% of the total timesteps. The target network is updated every 500 timesteps. We update $\theta$ using the gradient $\nabla_\theta \mathcal{L}(\theta)$, $\mathcal{L}(\theta) = (1 - \lambda)L^{\mathrm{main}}(\theta) + \lambda L^{\mathrm{aux}}(\theta)$, where $L^{\mathrm{main}}(\theta)$ is the primary loss and $L^{\mathrm{aux}}(\theta)$ is the auxiliary loss in Yang et al. (2019). The weight $\lambda$ is linearly scheduled from 0 to 1 over the first 75% and 25% percent of the total timesteps in traffic and LunarLandar, respectively. Sampling preference vectors $\omega_m \in \Delta^m$ during training and execution follows the uniform Dirichlet distribution.

For the three online dimension reduction methods (our approach, the autoencoder, and our implementation of online NPCA), we utilize the Adam optimizer for updates. In our method, the matrix $A$ is initialized with each entry set to $1/K$. The neural network $g_\phi$ has an input dimension of $m$, two hidden layers of 32 units each, and ReLU activations after each hidden layer. The output layer has a size of $K$. We use a dropout rate of 0.75 and 0.25 in in traffic and LunarLandar, respectively (with 0 meaning no dropout). Equation 11 is optimized with a learning rate of 0.0003 and an update interval of 5 timesteps.

For the autoencoder, the encoder network has an input size of $K$, two hidden layers with 32 units each, and ReLU activations after each hidden layer. The output layer has a size of $m$. The decoder follows the same architecture as $g_\phi$, but without dropout. The reward reconstruction loss is optimized with a learning rate of 0.0001 and an update interval of 20 timesteps.

For the online NPCA, we use ReLU parameterization for efficient learning (also implemented in PyTorch (Paszke et al., 2019)) to meet the constraint on matrix $U$. The matrix $U$ is initialized similarly with each entry set to $1/K$. NPCA is optimized with a learning rate of 0.0001, an update interval of 20(traffic)/50(LunarLander) timesteps, and $\beta = 50000(\mathrm{traffic})/1000(\mathrm{LunarLander})$. The reduced vector representation of $r$ is $U^T(r - \mu) \in \mathbb{R}^m$, following the PCA assumption that the transformed vectors are centered (Zass & Shashua, 2006; Cardot & Degras, 2018). For NPCA-ortho in traffic, increasing the value of $\beta$ did not yield better orthonormality, so we set the update interval to 5 timesteps, keeping the same $\beta$ value.

For incremental PCA, we recursively update the sample mean vector of rewards as $\mu_{t+1} = \frac{t}{t+1}\mu_t + \frac{1}{t+1}r_{t+1} \in \mathbb{R}^K$ and the sample covariance matrix as $C_{t+1} = \frac{t}{t+1}C_t + \frac{t}{(t+1)^2}(r_{t+1} - \mu_t)(r_{t+1} - \mu_t)^\top \in \mathbb{R}^{K \times K}$ for each timestep $t$ (Cardot & Degras, 2018). Every 20 timesteps, we eigen-decompose the covariance matrix, selecting the top $m$ eigenvectors $u_1, \ldots, u_m \in \mathbb{R}^K$ corresponding to the largest eigenvalues, and update $U = [u_1, \ldots, u_m] \in \mathbb{R}^{K \times m}$. The reduced vector representation of $r$ is $U^T(r - \mu) \in \mathbb{R}^m$, assuming the vectors are centered (Cardot & Degras, 2018). $U$ is initialized as a matrix with each entry set to $1/K$. For evaluation, we generated fifteen and thirty five equidistant points on the simplex for LunarLander and the traffic environment, respectively. For evenly distributed sampling and calculating hypervolume and sparsity, we use the implementation provided in Felten et al. (2023). We use infrastructures of Intel Xeon Gold 6238R CPU @ 2.20GHz and Intel Core i9-10900X CPU @ 3.70GHz.

