# OpenReview forum: "Reward Dimension Reduction for Scalable Multi-Objective Reinforcement Learning"
_ICLR.cc/2025/Conference — ICLR 2025 Poster_

### Official Review · Reviewer_ywhQ · 2024-10-27

**Soundness:** 3
**Presentation:** 3
**Contribution:** 2
**Rating:** 6
**Confidence:** 2

**Summary:**

This work studies multi objective RL (MORL), considering the Pareto frontier and measures hypervolume and sparsity of solutions to judge viability in downstream tasks. They propose a new dimensionality reduction technique to tackle existing challenges in high-dimensional problems. This can help to collapse similar objectives to reduce the effective reward space. The authors give some theoretical and empirical demonstration validating their method.

**Strengths:**

- Simplicity of idea and execution
- Providing well-founded theory to motivate their method, and proof given is easy to follow
- Modular (could be applied with other MORL methods)
- Provides potential new SOTA for the sample experiment
- nice explanation of MORL problem for non-experts
- I was hoping to see an ablation on dropout as soon as it was mentioned, and it was indeed given (Maybe mention Table 3 in L363)

**Weaknesses:**

Overall, I can't quite tell if I should think of this as a theory or experiment paper. For the former, I believe the theory given is not strong enough. But for the latter, there is only one environment, so I think it is not fair to claim total SOTA performance without a more systematic set of experiments. Below I'll give some points of weakness mixed in with related questions.

- Somewhat unclear empirics (the authors only provide scalar statistics on a single environment)
- Theory is somewhat weak. Specifically, they provide a sufficient, but not necessary condition. However, the surrounding text (Line 320) makes it sound like a necessary condition "we require...". Just be careful to not over-sell the result.
- I'm unsure how limiting the linearity in Theorem 1 is. Although it is beneficial for its simplicity from a computational perspective, is something being lost here? It seems to only capture linear relationships among various objectives. Can a counterexample be provided to better understand the breakdown of the (<=) other direction of the theorem?
- Some pre-assumed knowledge: As mentioned in L412, the final rank=4 was assumed to be known. But this would come at the end of training. How big an impact is there in different choices of rank? Can the rank be changed adaptively?
- L449: It's a bit unclear to me if NPCA was tuned appropriately for a fair comparison. Can better choices of the regularization strength be chosen? (Also, where's the reference to NPCA, I think I missed it)

I'm not totally aware of the MORL literature, but are there any other baselines with which to compare?

**Questions:**

Beyond those in the previous section, here are a few more questions for the authors. If the authors are able to sufficiently abate my concerns, I can raise the score.

- In your theorem, do you think it would be possible to generalize to PSD matrices? I'm not seeing the hurdle, but it would be enlightening (for me at least (:  )  to discuss either way
- Can you discuss any potential use of work like [3] to reduce reward space by canonicalizing objectives (ensuring they truly are distinct)? Would that be applicable here?
- RE Table 4, Can you also give an ablation of "-positivity"? (with the rowst constraint)
- L500 is unclear. Why does finite ep length introduce more overhead?
- Relatedly, what is the complexity of the constrained optimization problem?
- Table 1: why did you choose this reference point? Maybe you can normalize the values wrt ${10^{4*16}}=10^64$ then?
- Can you provide the learning curves or more dynamic information about your algorithm? How about other statistics (cf https://github.com/google-research/rliable)
- L325, A brief discussion on the connection to [1] would be interesting, if possible
- How does this work (or MORL more generally) relate to the successor features framework, e.g. [2]?

minor:
- Fig 2, can you change to e.g. $|AB|^2$ if that is meant. Also the quality of figures is a bit low, maybe just increase DPI a bit
- "works" -> "work"

[1]: https://proceedings.neurips.cc/paper_files/paper/2022/file/f600d1a3f6a63f782680031f3ce241a7-Paper-Conference.pdf
[2]: https://arxiv.org/abs/1606.05312
[3]: http://aima.cs.berkeley.edu/~russell/papers/iclr21-epic.pdf


I'm genuinely curious to understand the effect of dropout. I think the explanation starting at L475 is a bit confusing (for me). Can you offer any additional explanations/re-wordings?

---

> ### Author Response · Authors · 2024-11-25
>
> Thank you for your valuable comments on our work. We have uploaded a portion of our responses here and will address the remaining questions shortly (W2-3, Q1, Q3, and Q8-9).
>
> 1. **W1. "Somewhat unclear empirics (the authors only provide scalar statistics on a single environment)"**
>
> As suggested by Reviewer Pwfy, we have added another environment from existing MORL benchmarks even though it has fewer objectives than the traffic environment. This addition strengthens the validity of our experimental evaluation.
>
> The new environment, LunarLander-5D [1, 2], features a five-dimensional reward structure where the agent’s goal is to successfully land a lunar module on the moon's surface. Each reward dimension represents: (i) a sparse binary indicator for successful landing, (ii) a combined measure of the module's position, velocity, and orientation, (iii) the fuel cost of the main engine, (iv) the fuel cost of the side engines, and (v) a time penalty. This environment presents significant challenges because failing to balance these objectives effectively can easily lead to unsuccessful landings [1, 2].
>
> We evaluated our method using hypervolume and sparsity metrics, with all dimension reduction methods set to a tuned reduced dimension of $m=3$ and the reference point for hypervolume evaluation set to $(0, -150, -150, -150, -150) \in \mathbb{R}^{5}$.  The results show that our algorithm outperforms the baseline approaches in this environment as well.
>
> |                          | Base              | PCA               | AE                | NPCA              | **Ours**          |
> |--------------------------|-------------------|-------------------|-------------------|-------------------|-------------------|
> | **HV (× 10⁸, ↑)**        | 7.6 ± 9.0         | 8.8 ± 7.9         | 0                 | 4.9 ± 6.8         | **37.1** ± 6.7    |
> | **SP (× 10², ↓)**        | 31.2 ± 25.3       | 188.6 ± 180.7     | 31.3 ± 30.6       | 53.0 ± 47.7       | **1.1** ± 1.2     |
>
> &nbsp;
>
> 2. **W4-5. Effect of reduced dimension $m$ + NPCA**
>
> $m$ is a hyperparameter for our algorithm, and selecting an appropriate value in practice is achievable. This is because we can estimate the effective rank of the sample covariance matrix recursively [3] during the early exploration phase of the RL algorithm, rather than throughout the entire training process. We found that this straightforward approach performs effectively in our experiments.
>
>
> The following table shows the impact of varying the reduced dimensionality $m$ in our method in the traffic environment. As $m$ increases from 4 to 6, sparsity rises significantly while hypervolume remains constant. This results in situations where only a few objectives perform well. When $m$ increases from 8 to 10, both sparsity increases and hypervolume decreases, leading to a lower-quality set of returns in the original reward space. Based on these observations, we set $m=4$ for our traffic environment experiments.
>
> | $m$                     | 2        | 4        | 6        | 8        | 10       |
> |-------------------------|----------|----------|----------|----------|----------|
> | **HV (× 10⁶³, ↑)**      | 1.4      | 1.7      | 1.7      | 1.9      | 1.1      |
> | **SP (× 10⁵, ↓)**       | 1.5      | 2.3      | 30       | 20       | 81       |
>
> Regarding NPCA, while it was originally designed for batch data in [4], we adapted it for online learning. If we were to directly apply the formulation from [4], we would need an additional positivity constraint $U \geq 0$. This constraint would introduce an extra hyperparameter, potentially reducing overall stability. To ensure a fair comparison with our method, which uses direct parameterization, we implemented an online variant of NPCA that also adopts direct parameterization to satisfy  $U \geq 0$. Additionally, we fine-tuned key hyperparameters such as $\beta$, the learning rate for gradient descent, and the update interval for NPCA.
>
> &nbsp;
>
> 3. **Q2. "Can you discuss any potential use of work like 3 to reduce reward space by canonicalizing objectives?"**
>
> The EPIC distance is a pseudometric that measures the "distance" between reward functions by canonicalizing them. It is widely used when the true reward function is unavailable during the training phase, such as in [5], where vision-language models are used for reward learning. The EPIC distance decreases as the correlation between two canonicalized reward functions increases.
>
> One could canonicalize each objective in MORL and combine some rewards based on their closest canonicalized functions. However, there are two key challenges with this approach. (i) Calculating the EPIC distance efficiently in complex RL environments is non-trivial. (ii) Designing a method to combine (or cluster) objectives is another challenge. For instance, determining the mixing ratio of the reward functions requires careful consideration. Exploring this direction further represents a promising avenue for extending our work.

---

> ### Author Response · Authors · 2024-11-25
>
> 4. **Q4-5. "Why does finite ep length introduce more overhead?" + complexity**
>
> We apologize for any confusion caused by our earlier explanation. Introducing a bias term $b$ does not increase computational overhead. Our point was that adding this term does not provide any additional benefit in our learning process, as mentioned in lines 501-502. Overall, our method is simple and modular as you mentioned in Strengths of our work, ensuring that it does not introduce significant computational burden.
>
> &nbsp;
>
> 5. **Q6-7. "Why did you choose this reference point?" + robust statistics**
>
> We selected the reference point based on observations that, after the initial exploration phase, most points in the current Pareto fronts fell within this defined region (except for PCA). To enhance the statistical reliability of our experimental results, we applied a 12.5% trimmed mean by excluding the seeds with maximum and minimum hypervolume values over eight random seeds and reporting the averages of the metrics over the remaining six random seeds. This method is similar to the interquartile mean (IQM) and offers robustness against outliers [6] while maintaining more seeds (i.e., six versus four), balancing the standard average and IQM. The updated table shows that **our algorithm consistently outperforms the baseline methods** in the traffic environment in terms of both IQM and 12.5% trimmed mean. For better visualization, here we report hypervolume scaled by $10^{61}$ and sparsity by $10^5$.
>
> | IQM                  | Base              | PCA               | AE                | NPCA              | **Ours**          |
> |--------------------------|-------------------|-------------------|-------------------|-------------------|-------------------|
> | **HV (× 10⁶¹, ↑)**       | 2.7 ± 5.1         | 0                 | 0    | 17.9 ± 10.3       | **170.2** ± 35.6  |
> | **SP (× 10⁵, ↓)**        | 1160 ± 569       | 3828 ± 1826       | 6684 ± 686       | 47.9 ± 61.5       | **2.0** ± 1.1   |
>
> | 12.5% Trimmed                  | Base              | PCA               | AE                | NPCA              | **Ours**          |
> |--------------------------|-------------------|-------------------|-------------------|-------------------|-------------------|
> | **HV (× 10⁶¹, ↑)**       | 4.4 ± 6.8         | 0                 | 0.007 ± 0.018     | 19.4 ± 15.3       | **166.9** ± 48.1  |
> | **SP (× 10⁵, ↓)**        | 1842 ± 1290       | 3837 ± 2164       | 7834 ± 3323       | 34.2 ± 52.3       | **2.3** ± 1.0   |
>
> Additionally, we periodically evaluated the hypervolume of our method and NPCA, dividing the training process into five segments. Our results indicate that our method learns faster compared to NPCA. We omitted other baselines from this analysis as their hypervolume values were relatively small in comparison.
>
> | **HV (× 10⁶², ↑)**       | 1/5              | 2/5               | 3/5               | 4/5               | 5/5               |
> |--------------------------|------------------|-------------------|-------------------|-------------------|-------------------|
> | **Ours**                 | 0.427 ± 1.044    | 11.60 ± 6.57      | 14.33 ± 8.42      | 18.00 ± 5.73      | 16.69 ± 4.81      |
> | **NPCA**                 | 0                | 0.004 ± 0.011     | 0.148 ± 0.193     | 0.043 ± 0.078     | 1.940 ± 1.53      |
>
> For completeness, we also present hypervolume values for different reference points in Appendix B.2 in our pdf, demonstrating that our algorithm consistently outperforms the baseline methods.
>
> &nbsp;
>
> 6. **Regarding visualization in high-dimensional space**
>
> As suggested by Reviewer Pwfy, we visualized the Pareto frontier obtained in the traffic environment for each algorithm using t-SNE. The corresponding figure is shown in Appendix B.1 in our pdf. We emphasize that our primary objective is to **cover a broad region of the true Pareto frontier, not merely to cover a wide region of a high-dimensional reward space itself**. Although AE solutions may appear widely distributed, this does not necessarily imply extensive coverage of the true Pareto frontier because the true Pareto frontier is a subset of the original space. Given that AE yields a low hypervolume, it is less likely to represent a wide range of the true Pareto frontier.
>
> According to the overlap analysis, smaller overlaps with the Base method suggest a different local structure, for example, either in a way of our method (with high hypervolume) or a way of PCA (with very low hypervolume). This qualitative difference suggests that our method and PCA are distinct in their approach to exploring the solution space. Also, NPCA overlaps with more points from the Base and AE methods than our method, demonstrating the insufficiency of NPCA in covering a larger true Pareto frontier than the Base method.

---

> ### Author Response · Authors · 2024-11-26
> **Additional reply for clarification**
>
> Dear Reviewer ywhQ,
>
> We here address the remaining questions (W2-3, Q1, Q3, Q8-9, and dropout) as follows.
>
> 7. **Q1 and Q3. "Generalization to PSD matrices" + "An ablation of -positivity"**
>
> The following table shows our updated ablation study examining the impact of different conditions on the dimension reduction function $f$. We observed that if we remove the positivity condition while maintaining the row-stochasticity constraint, the algorithm produces zero hypervolume. This is due to the lack of the positivity condition required by Theorem 1.
>
> Therefore, the positivity condition is essential for maintaining Pareto-optimality, while the row-stochasticity constraint improves the efficiency of online learning under the positivity condition. Since PSD matrices do not necessarily guarantee that all elements are positive, extending Theorem 1 to PSD matrices presents a non-trivial challenge. (If we did not catch what you meant, please notify us.)
>
> |                              | **Ours**       | +bias            | -rowst           | -positivity       | -rowst, -positivity |
> |------------------------------|----------------|------------------|------------------|-------------------|----------------------|
> | **HV (× 10⁶¹, ↑)**           | **166.9**      | 132.9            | 46.8             | 0                 | 0                    |
> | **SP (× 10⁵, ↓)**            | **2.3**        | 2.7              | 38.8             | 4066.6            | 5310.7               |
>
> &nbsp;
>
> 8. **W2-3. Discussion on Theorem 1**
>
> First, we revise the sentence in Line 320 as follows: "In short, the condition of $f(r) = Ar + b$ with $A \in \mathbb{R}^{m \times K}_+$ guarantees the preservation of Pareto-optimality in equation 7."
>
> Next, we acknowledge that theoretically analyzing the opposite direction of Theorem 1 is challenging. To find conditions for a counterexample of $f$ beyond $f(r) = Ar + b$ with $A \in \mathbb{R}^{m \times K}_+$, we may follow a similar flow in the proof of the Theorem 1. Given $\omega_m \in \Delta^m$, suppose $\exists \pi' \in \Pi$ s.t.
>
> $\mathbb{E}_{\pi'} \left[ \sum_t \gamma^t r_t \right]$
>
> Pareto dominates $\mathbb{E}_{\pi^*_m(\cdot|\cdot, \omega_m)} \left[ \sum_t \gamma^t r_t \right]$ in the original reward space.
>
> We first impose that $f = [f_1, \cdots, f_m]$ be a *monotonically strictly increasing function* satisfying $A >_P B \Rightarrow f_j(A) > f_j(B)$ for all $1 \leq j \leq m$ [1].
>
> Then we have $f(\mathbb{E}_{\pi'} \left[ \sum_t \gamma^t r_t \right])$
>
> Pareto dominates $f(\mathbb{E}_{\pi^*_m(\cdot|\cdot, \omega_m)} \left[ \sum_t \gamma^t r_t \right])$.
>
> If $f$ further satisfies $\mathbb{E}_{\pi'} \left[ \sum_t \gamma^t f(r_t) \right]$
>
> Pareto dominates $\mathbb{E}_{\pi^*_m(\cdot|\cdot, \omega_m)} \left[ \sum_t \gamma^t f(r_t) \right]$ (which we denote as Condition 2 here), this gives a contradiction and $f$ becomes our target counterexample. This is directly satisfied when $f$ is affine. However, it is difficult to find such an example, primarily due to the inequality in $>_P$.
>
> Alternatively, we conducted an empirical analysis by relaxing Condition 2 and directly optimizing $f$ under the sole constraint of a strictly monotonically increasing function. As part of our ablation study, we parameterized $f$ as a strictly monotonically increasing function using a neural network (similar to approaches like [7] but maintaining strict monotonicity) and trained it within the traffic environment, denoted by "monotone."
>
> |                              | Monotone | **Ours**   |
> |------------------------------|----------|------------|
> | **HV ($ \times 10^{61}, \uparrow $)** | 0        | **166.9**   |
> | **SP ($ \times 10^{5}, \downarrow $)**| 5353.0   | **2.3**     |
>
> Our findings revealed that this approach resulted in a hypervolume of **zero**, similar to the "-positivity" and "-rowst, -positivity" cases. This suggests that merely imposing a strictly monotonically increasing function condition is insufficient to construct a meaningful counterexample in practice. Importantly, nonzero hypervolume was only achieved when both the affine and positivity conditions were satisfied, as demonstrated in the "-rowst" case from the previous table. These results underscore the empirical effectiveness of our algorithm based on Theorem 1. (If you have further suggestions, we would be happy to discuss them!)

---

> ### Author Response · Authors · 2024-11-26
> **Additional reply for clarification**
>
> 9. **Q8-9. Regarding additional references and baselines**
>
> We thank the reviewer for suggesting interesting papers.
>
> (1) Regarding paper "Successor Features for Transfer in RL" (Barreto et al., 2018):
>
> A seminal survey [8] in MORL provides a comprehensive comparison between MORL and successor features (SF), where a scalar reward is decomposed into a product of state features and task weights.
>
> State features and task/goal weights in SF can be viewed analogously to the multi-objective reward and linear weight vectors. However, according to [8], SF does not directly observe individual objectives; it operates with a scalar reward function. In some cases, constructing a scalar reward first and inferring it back into multiple objectives can be sub-optimal in conventional MORL settings. However, SF offers performance guarantees in transfer learning scenarios, which is advantageous depending on the context.
>
> (2) Regarding paper "Exploit Reward Shifting in Value-Based Deep-RL" (Sun et al., 2022):
>
> This paper investigates how linear reward shifting affects Q-network initialization. Positive shifts promote conservative behavior, while negative shifts encourage exploration. Although both studies consider linear transformations, there are key differences.
>
> (i) Our work focuses on approximating the true Pareto frontier in a multi-objective space, while this paper examines behavior in standard RL. Consequently, our approach only considers positive shifts to preserve Pareto-optimality, whereas their findings also highlight the effectiveness of negative shifts in online RL settings.
> (ii) While their analysis is based on Q-function approximation, our method is not restricted to function approximation cases.
>
> Combining insights from both studies—such as applying linear transformations in offline MORL within high-dimensional spaces—could lead to interesting future developments.
>
> Unlike our modular dimension reduction approach, which is compatible with any MORL algorithm, most existing dimension reduction techniques are designed for static (batch-based) datasets. As discussed in the Related Work section, only a few methods are suitable for online settings. Incremental PCA and online autoencoders are commonly used for online dimension reduction [3, 9]. We implemented an online version of NPCA in this work, but developing online variants of other dimension reduction methods remains challenging [10]. Therefore, we compared our algorithm with the online version of PCA, AE, and NPCA.
>
> &nbsp;
>
> 10. **Clarification on the effect of dropout**
>
> We clarify the effect of dropout by considering the quality of the return vector over the training. As mentioned earlier, our goal is to cover a broad region of the true Pareto frontier, rather than simply expanding coverage in the high-dimensional reward space. In the early stages of training, the collected return data are less likely to lie on the true Pareto frontier. As learning progresses, higher-quality data are gathered. Dropout helps to leverage this improved data by intentionally slowing down the fitting process of the function $f$, thereby preventing premature convergence to suboptimal solutions.
>
> &nbsp;
>
> References
>
> [1] Felten et al., "A toolkit for reliable benchmarking and research in multi-objective reinforcement learning", 2023.
>
> [2] Hung et al., "Q-Pensieve: Boosting Sample Efficiency of Multi-Objective RL Through Memory Sharing of Q-Snapshots", 2023.
>
> [3] Herv´e Cardot and David Degras, "Online principal component analysis in high dimension: Which algorithm to choose?", 2018.
>
> [4] Ron Zass and Amnon Shashua, "Nonnegative sparse pca", 2006.
>
> [5] Rocamonde et al., "Vision-Language Models are Zero-Shot Reward Models for Reinforcement Learning", 2024.
>
> [6] Maronna et al., "Robust Statistics: Theory and Methods", 2006.
>
> [7] Rashid et al., "QMIX: Monotonic Value Function Factorisation for Deep Multi-Agent Reinforcement Learning", 2018.
>
> [8] Hayes et al., "A practical guide to multi-objective reinforcement learning and planning", 2022.
>
> [9] Bank et al., "Autoencoders", 2020.
>
> [10] Leland McInnes and John Healy, "UMAP: uniform manifold approximation and projection for dimension reduction", 2018.

---

> > ### Author Response · Authors · 2024-11-26
> > **Additional reply for clarification**
> >
> > Finally, the minor suggestions will be reflected in our final version.
> >
> > We sincerely thank the reviewer for the detailed comments that significantly helped improve the quality of our work!

---

> > > ### Author Response · Authors · 2024-11-29
> > > **Thank you for the comments**
> > >
> > > Dear Reviewer ywhQ,
> > >
> > > Thank you for your detailed and insightful comments, which have greatly contributed to improving our paper.
> > >
> > > Could you kindly review our responses to ensure we have addressed your questions thoroughly?
> > >
> > > We appreciate your time and effort!

---

> > > > ### Comment · Reviewer_ywhQ · 2024-11-30
> > > > **Response**
> > > >
> > > > Apologies for the delay -- I believe the authors have mostly addressed my concerns. Thanks for adding another environment and answering my questions. I also appreciate the discussion in the context of the references you mentioned. There is just a quick question:
> > > >
> > > > "Dropout helps to leverage this improved data by intentionally slowing down the fitting process of the function, thereby preventing premature convergence to suboptimal solutions." Naively, there are surely other ways of slowing the fitting process like just reducing learning rate or udpate frequency. Maybe there is something more nuanced here going on with dropout specifically? Apparently it can be seen as equivalent to a regularization: [1] or sec 3 in [2].
> > > >
> > > > [1]: https://proceedings.neurips.cc/paper/2013/file/38db3aed920cf82ab059bfccbd02be6a-Paper.pdf
> > > > [2]: https://nlp.stanford.edu/pubs/sidaw13fast.pdf

---

> ### Author Response · Authors · 2024-12-01
>
> Thank you for your insightful question and for helping us better understand the deeper nature of dropout!
>
>
> Dropout can be viewed as a form of regularization approximately equivalent to applying an L2 penalty after normalizing the input vectors [1]. L2 regularization helps prevent scenarios where certain weights in a neural network become excessively large, causing the network to prioritize specific features disproportionately. It is particularly effective when dealing with high feature correlations [2]. Additionally, dropout enables more sophisticated regularization than standard L2 regularization, enhancing the model's generalization capabilities [1].
>
> In our context, each output feature of the reconstructor $g_\phi$ corresponds to an objective in MORL, and implicit correlations exist among these objectives. By applying dropout, we mitigate the risk of the model prematurely focusing on a subset of the $K$ objectives by suppressing the weights of $g_\phi$ to be overly large. This can explain why sparsity is increased when we do not use dropout (as seen in Table 4 in our pdf): we may lose the opportunity for diverse solutions on the Pareto frontier due to the premature focus of our learning on a subset of the $K$ objectives without the weight regularization.
>
> We will incorporate this explanation in the final version.
>
> Thank you once again for your valuable feedback.
>
>
> &nbsp;
>
> References
>
> [1] Wager et al., "Dropout Training as Adaptive Regularization".
>
> [2] https://spotintelligence.com/2023/05/26/l1-l2-regularization/

---

> > ### Comment · Reviewer_ywhQ · 2024-12-01
> > **Response**
> >
> > Great - thank you for the quick response and in-depth discussion with me and the other reviewers, it was helpful in forming my understanding. Given that I am not an expert in the MORL field, I cannot drastically improve the score, but I will improve it to reflect these discussions and useful additions to the manuscript.

---

> > > ### Author Response · Authors · 2024-12-01
> > >
> > > Thank you so much for your valuable comments again. We appreciate your time and efforts.

---

### Official Review · Reviewer_iWoJ · 2024-10-28

**Soundness:** 2
**Presentation:** 2
**Contribution:** 2
**Rating:** 6
**Confidence:** 2

**Summary:**

This paper presents a reward reduction approach for multi-objective reinforcement learning (MORL) that transforms a high-dimensional reward vector into a lower-dimensional one. Unlike previous methods that primarily rely on static datasets, this approach supports online learning while ensuring that Pareto optimality is preserved after the transformation. Additionally, the authors introduce a comprehensive training and evaluation framework for reward dimension reduction in MORL.

**Strengths:**

1. This paper focuses on the important problem of multi-objective reinforcement learning in the presence of high-dimensional rewards.

**Weaknesses:**

1. This paper proposes applying an affine transformation to reduce the original high-dimensional reward vector to a lower-dimensional representation. However, the authors do not provide a theoretical justification demonstrating why reducing the reward dimension would improve multi-objective training, particularly when compression may result in information loss.
2. One major limitation is the experimental scope. The experiments are conducted solely on a traffic light control environment, without exploring the impact of varying the dimensionality mmm, which undermines the generalizability and robustness of the results.

**Questions:**

1. In Tables 1 and 2, it seems the scale of the reported metrics is really large. Could the author illustrate more about metrics?

---

> ### Author Response · Authors · 2024-11-25
>
> Thank you for your valuable comments on our work. We appreciate your feedback and kindly ask you to confirm whether our responses address your concerns.
>
>
> 1. **"Could the author illustrate more about metrics?" + Illustration in our high-dimensional reward space**
>
> The hypervolume (HV) metric measures the volume in the objective space dominated  by the current set of Pareto frontier points, bounded by a reference point. It provides a scalar value that quantifies how well the policies cover the objective space. The reported metric values are large because the hypervolume increases exponentially with the number of objectives [1].
>
> Sparsity (SP) evaluates how well the policies are distributed within the objective space. A low sparsity value indicates that the Pareto frontier points are evenly distributed, offering a diverse range of trade-offs among the objectives. Thus, sparsity serves as an indicator of how uniformly the set of return vectors is spread. These two metrics are widely recognized in the MORL community [2].
>
> We also visualized the Pareto frontier obtained in the traffic environment for each algorithm using t-SNE. The corresponding figure has been added to the **(newly created) supplementary material**.
>
> We emphasize that our primary objective is to **cover a broad region of the true Pareto frontier, not merely to cover a wide region of a high-dimensional reward space itself**. Although AE solutions may appear widely distributed, this does not necessarily imply extensive coverage of the true Pareto frontier because the true Pareto frontier is a subset of the original space. Given that AE yields a low hypervolume, it is less likely to represent a wide range of the true Pareto frontier.
>
> According to the overlap analysis, smaller overlaps with the Base method suggest a different local structure, for example, either in a way of our method (with high hypervolume) or a way of PCA (with very low hypervolume). This qualitative difference suggests that our method and PCA are distinct in their approach to exploring the solution space. Also, NPCA overlaps with more points from the Base and AE methods than our method, demonstrating the insufficiency of NPCA in covering a larger true Pareto frontier than the Base method.
>
> &nbsp;
>
> 2. **Regarding exploring the impact of varying the dimensionality**
>
> The following table shows the impact of varying the reduced dimensionality $m$ in our method in the traffic environment. As $m$ increases from 4 to 6, sparsity rises significantly while hypervolume remains constant. This results in situations where only a few objectives perform well [2]. When $m$ increases from 8 to 10, both sparsity increases and hypervolume decreases, leading to a lower-quality set of returns in the original reward space. Based on these observations, we set $m=4$ for our traffic environment experiments.
>
> | $m$                     | 2        | 4        | 6        | 8        | 10       |
> |-------------------------|----------|----------|----------|----------|----------|
> | **HV (× 10⁶³, ↑)**      | 1.4      | 1.7      | 1.7      | 1.9      | 1.1      |
> | **SP (× 10⁵, ↓)**       | 1.5      | 2.3      | 30       | 20       | 81       |
>
> &nbsp;
>
> 3. **"The experiments are conducted solely on a traffic light control environment."**
>
> As suggested by Reviewer Pwfy, we have added another environment from existing MORL benchmarks even though it has fewer objectives than the traffic environment. This addition strengthens the validity of our experimental evaluation.
>
> The new environment, LunarLander-5D [3, 4], features a five-dimensional reward structure where the agent’s goal is to successfully land a lunar module on the moon's surface. Each reward dimension represents: (i) a sparse binary indicator for successful landing, (ii) a combined measure of the module's position, velocity, and orientation, (iii) the fuel cost of the main engine, (iv) the fuel cost of the side engines, and (v) a time penalty. This environment presents significant challenges because failing to balance these objectives effectively can easily lead to unsuccessful landings [3, 4].
>
> We evaluated our method using hypervolume and sparsity metrics, with all dimension reduction methods set to a tuned reduced dimension of $m=3$ and the reference point for hypervolume evaluation set to $(0, -150, -150, -150, -150) \in \mathbb{R}^{5}$. The results show that **our algorithm outperforms the baseline approaches in this environment as well**.
>
> |                          | Base              | PCA               | AE                | NPCA              | **Ours**          |
> |--------------------------|-------------------|-------------------|-------------------|-------------------|-------------------|
> | **HV (× 10⁸, ↑)**        | 7.6 ± 9.0         | 8.8 ± 7.9         | 0                 | 4.9 ± 6.8         | **37.1** ± 6.7    |
> | **SP (× 10², ↓)**        | 31.2 ± 25.3       | 188.6 ± 180.7     | 31.3 ± 30.6       | 53.0 ± 47.7       | **1.1** ± 1.2     |

---

> ### Author Response · Authors · 2024-11-25
>
> 4. **"The authors do not provide a theoretical justification demonstrating why reducing the reward dimension would improve multi-objective training."**
>
> The challenge in MORL lies in effectively covering all possible preferences during training. In most previous MORL algorithms, agents sample random preferences in each episode to collect diverse behaviors. However, performing this sampling naively in high-dimensional spaces becomes computationally expensive because the coverage (or hypervolume) grows exponentially with the number of objectives [1]. To address this, our approach integrates dimension reduction techniques to down the search space while preserving the most relevant information.
>
> Dimension reduction techniques are widely applied in various machine learning domains. When the reduced dimension $m$ is chosen appropriately, the most significant features of high-dimensional data are preserved while irrelevant noise is filtered out [5, 6]. Selecting a suitable $m$ in practice is feasible because we can estimate the effective rank of the sample covariance matrix $C_t$ during the early exploration phase of the RL algorithm (not throughout the entire training process). This is done by recursively updating the sample mean vector of rewards: $\mu_{t+1}  = \frac{t}{t+1}\mu_{t} + \frac{1}{t+1}r_{t+1} \in \mathbb{R}^K$ and the covariance matrix: $C_{t+1} = \frac{t}{t+1} C_{t} + \frac{t}{(t+1)^2} (r_{t+1} - \mu_t)(r_{t+1} - \mu_t)^\top \in \mathbb{R}^{K \times K}$ for each timestep $t$ [5].
>
> Moreover, we have **theoretically demonstrated in Theorem 1 that our dimension reduction method preserves Pareto-optimality** in equation 7. This theoretical justification explains why our approach enhances multi-objective training compared to other methods that do not guarantee Pareto-optimality preservation.
>
> If you have further questions, please feel free to ask. :)
>
> &nbsp;
>
> References
>
> [1] Weijia Wang and Michèle Sebag, "Hypervolume indicator and dominance reward based multi-objective Monte-Carlo Tree Search", 2013.
>
> [2] Hayes et al., "A practical guide to multi-objective reinforcement learning and planning", 2022.
>
> [3] Felten et al., "A toolkit for reliable benchmarking and research in multi-objective reinforcement learning", 2023.
>
> [4] Hung et al., "Q-Pensieve: Boosting Sample Efficiency of Multi-Objective RL Through Memory Sharing of Q-Snapshots", 2023.
>
> [5] Herv´e Cardot and David Degras, "Online principal component analysis in high dimension: Which algorithm to choose?", 2018.
>
> [6] Bank et al., "Autoencoders", 2020.

---

> > ### Comment · Reviewer_iWoJ · 2024-11-25
> >
> > Thank you for the response. It effectively addresses my concern, and I will adjust the score accordingly.

---

> > > ### Author Response · Authors · 2024-11-26
> > >
> > > We thank the reviewer for raising the score accordingly, and we also appreciate the insightful comments that helped us improve our work!

---

### Official Review · Reviewer_uQAS · 2024-10-30

**Soundness:** 3
**Presentation:** 4
**Contribution:** 3
**Rating:** 8
**Confidence:** 3

**Summary:**

The authors propose a novel method for reward dimensionality reduction as well as a new evaluation protocol & benchmark for high dimensional multi objective reinforcement learning. The developed approach reduces the dimensionality of the rewards with an affine transformation that is additionally constrained to be row stochastic & feature only positive elements in its matrix. The authors show in an empirical evaluation that their approach beats relevant baselines in a high-dimensional multi-objective environment.

**Strengths:**

To the best of my knowledge, the approach is novel. It is furthermore simple & elegant and leads to demonstrated improvements over baselines on a high-dimensional multi-objective environment. In addition, the paper proposes a new evaluation setup for their work. Finally, it is well written & understandable also for readers outside of the multi-objective sphere.

**Weaknesses:**

The empirical evaluation is limited to a single environment. This is partially due to the fact that the authors address a problem that has so far been only explored to a limited degree before, however it still means the demonstrated superiority of the approach has to be taken with a grain of salt. Also, the evaluation doesn't feature uncertainty estimates. I would be glad to improve my score if those two points are addressed.

**Questions:**

I am not quite sure whether due to the affine nature of your approach, the resulting transformation function will deterministically always be the same given the same data (?), but I presume that the online data collection as well as the base RL algorithm used will introduce some stochasticity into the results, so could you please add corresponding uncertainty estimates in your tables (it seems you should have the data already anyways since you mention 8 random seeds)?

I think the following paper could enrich your related work section, would you care to review and potentially add it?
Weber, M., Swazinna, P., Hein, D., Udluft, S., & Sterzing, V. Learning Control Policies for Variable Objectives from Offline Data. In 2023 IEEE Symposium Series on Computational Intelligence (SSCI) (pp. 1674-1681). IEEE.

---

> ### Author Response · Authors · 2024-11-25
>
> Thank you for your valuable comments on our work. We appreciate your feedback and kindly ask you to confirm whether our responses address your concerns.
>
>
> 1. **"Could you please add corresponding uncertainty estimates in your tables?"**
>
> To improve the statistical reliability of our experimental results, we have included standard deviation values for each algorithm, incorporating uncertainty estimates based on an increased number of test preferences (as suggested by Reviewer Pwfy). The updated table demonstrates that **our algorithm consistently outperforms the baseline methods in the traffic environment**.
>
> |                          | Base              | PCA               | AE                | NPCA              | **Ours**          |
> |--------------------------|-------------------|-------------------|-------------------|-------------------|-------------------|
> | **HV (× 10⁶¹, ↑)**       | 4.4 ± 6.8         | 0                 | 0.007 ± 0.018     | 19.4 ± 15.3       | **166.9** ± 48.1  |
> | **SP (× 10⁵, ↓)**        | 1842 ± 1290       | 3837 ± 2164       | 7834 ± 3323       | 34.2 ± 52.3       | **2.3** ± 1.0   |
>
> &nbsp;
>
> 2. **"The empirical evaluation is limited to a single environment."**
>
> As suggested by Reviewer Pwfy, we have added another environment from existing MORL benchmarks even though it has fewer objectives than the traffic environment. This addition strengthens the validity of our experimental evaluation.
>
> The new environment, LunarLander-5D [1, 2], features a five-dimensional reward structure where the agent’s goal is to successfully land a lunar module on the moon's surface. Each reward dimension represents: (i) a sparse binary indicator for successful landing, (ii) a combined measure of the module's position, velocity, and orientation, (iii) the fuel cost of the main engine, (iv) the fuel cost of the side engines, and (v) a time penalty. This environment presents significant challenges because failing to balance these objectives effectively can easily lead to unsuccessful landings [1, 2].
>
> We evaluated our method using hypervolume and sparsity metrics, with all dimension reduction methods set to a tuned reduced dimension of $m=3$ and the reference point for hypervolume evaluation set to $(0, -150, -150, -150, -150) \in \mathbb{R}^{5}$. The results show that **our algorithm outperforms the baseline approaches in this environment as well**.
>
> |                          | Base              | PCA               | AE                | NPCA              | **Ours**          |
> |--------------------------|-------------------|-------------------|-------------------|-------------------|-------------------|
> | **HV (× 10⁸, ↑)**        | 7.6 ± 9.0         | 8.8 ± 7.9         | 0                 | 4.9 ± 6.8         | **37.1** ± 6.7    |
> | **SP (× 10², ↓)**        | 31.2 ± 25.3       | 188.6 ± 180.7     | 31.3 ± 30.6       | 53.0 ± 47.7       | **1.1** ± 1.2     |
>
> &nbsp;
>
> 3. **Regarding visualization in our high-dimensional reward space**
>
> As suggested by Reviewer Pwfy, we visualized the Pareto frontier obtained in the traffic environment for each algorithm using t-SNE. The corresponding figure has been added to the **(newly created) supplementary material**.
>
> We emphasize that our primary objective is to **cover a broad region of the true Pareto frontier, not merely to cover a wide region of a high-dimensional reward space itself**. Although AE solutions may appear widely distributed, this does not necessarily imply extensive coverage of the true Pareto frontier because the true Pareto frontier is a subset of the original space. Given that AE yields a low hypervolume, it is less likely to represent a wide range of the true Pareto frontier.
>
> According to the overlap analysis, smaller overlaps with the Base method suggest a different local structure, for example, either in a way of our method (with high hypervolume) or a way of PCA (with very low hypervolume). This qualitative difference suggests that our method and PCA are distinct in their approach to exploring the solution space. Also, NPCA overlaps with more points from the Base and AE methods than our method, demonstrating the insufficiency of NPCA in covering a larger true Pareto frontier than the Base method.
>
> &nbsp;
>
> 4. **Regarding paper citation**
>
> We are happy to add the paper you mentioned in our related work section. MORL addresses the situation where the true scalar reward function is not known in advance, which aligns with the recommended paper's motivation. We will add the paper to the final version of our draft. Thank you for your recommendation!
>
> &nbsp;
>
> References
>
> [1] Felten et al., "A toolkit for reliable benchmarking and research in multi-objective reinforcement learning", 2023.
>
> [2] Hung et al., "Q-Pensieve: Boosting Sample Efficiency of Multi-Objective RL Through Memory Sharing of Q-Snapshots", 2023.

---

> > ### Comment · Reviewer_uQAS · 2024-11-27
> >
> > I thank the authors for addressing all my concerns in detail. I will adjust my score as accordingly

---

> > > ### Author Response · Authors · 2024-11-27
> > >
> > > We sincerely thank the reviewer for raising the score to 8!
> > >
> > > The detailed comments helped us significantly improve the quality of our work.

---

### Official Review · Reviewer_Pwfy · 2024-11-01

**Soundness:** 3
**Presentation:** 3
**Contribution:** 3
**Rating:** 8
**Confidence:** 4

**Summary:**

This paper proposes a reward dimension reduction technique to improve the scalability of multi-objective RL (MORL). Specifically, it adopts an affine transformation and theoretically establishes conditions to guarantee the Pareto-optimality of learned policies in the original reward space and to build a valid mapping between the reduced and original preference spaces. Additionally, a reconstruction neural network helps preserve the information of the original rewards. The paper evaluates the proposed method through experiments with sixteen objectives, demonstrating superior performance compared to existing reduction and non-reduction methods.

**Strengths:**

- The paper is well-motivated. The scalability of MORL is crucial for real-world applications but is rarely studied. This work thoroughly considers the nature of multi-objective settings and develops a simple yet effective dimension reduction solution.
- The paper is well-written and well-organized, with clear theoretical explanations and implementation details that make the methodology easy to understand.
- Empirical evaluations and ablation studies are convincing, demonstrating the effectiveness of the proposed approach.

**Weaknesses:**

- The number of test preferences $N_e $, seems slightly insufficient, even for a 4-dimensional reduced reward space. I’m a bit concerned about whether this is enough to represent the performance across the whole Pareto front in the original 16-dim reward space.
- I wonder if it would be helpful to evaluate the dimension reduction technique on existing MORL benchmarks, even those with fewer objectives. Doing so could help assess the influence of the reduction method on performance and visualize the Pareto front in both the reduced and original reward spaces, helping us understand the relationship of Pareto in two reward spaces.
- In certain cases, learned policies may cover a broad Pareto front in the reduced reward space but occupy only a narrow area in the original reward space and lead to low sparsity. How could this situation be avoided or detected during evaluation? Without visualizing the high-dimensional Pareto front, I am curious whether metrics like hypervolume and sparsity are sufficient to detect such issues.

**Questions:**

See weaknesses

---

> ### Author Response · Authors · 2024-11-25
>
> Thank you for your valuable comments on our work. We appreciate your feedback and kindly ask you to confirm whether our responses address your concerns.
>
>
> 1. **"I wonder if it would be helpful to evaluate the dimension reduction technique on existing MORL benchmarks."**
>
> We have added an additional environment widely recognized in the MORL community to strengthen the validity of our experimental evaluation. The new environment, LunarLander-5D [1, 2], features a five-dimensional reward structure where the agent’s goal is to successfully land a lunar module on the moon's surface. Each reward dimension represents: (i) a sparse binary indicator for successful landing, (ii) a combined measure of the module's position, velocity, and orientation, (iii) the fuel cost of the main engine, (iv) the fuel cost of the side engines, and (v) a time penalty. This environment presents significant challenges because failing to balance these objectives effectively can easily lead to unsuccessful landings [1, 2]. \
> We evaluated our method using hypervolume and sparsity metrics, with all dimension reduction methods set to a tuned reduced dimension of $m=3$ and the reference point for hypervolume evaluation set to $(0, -150, -150, -150, -150) \in \mathbb{R}^{5}$.  We have incorporated standard deviation values for each algorithm to enhance statistical reliability, as suggested by Reviewer uQAS. The results show that **our algorithm outperforms the baseline approaches in this environment as well**.
>
> |                          | Base              | PCA               | AE                | NPCA              | **Ours**          |
> |--------------------------|-------------------|-------------------|-------------------|-------------------|-------------------|
> | **HV (× 10⁸, ↑)**        | 7.6 ± 9.0         | 8.8 ± 7.9         | 0                 | 4.9 ± 6.8         | **37.1** ± 6.7    |
> | **SP (× 10², ↓)**        | 31.2 ± 25.3       | 188.6 ± 180.7     | 31.3 ± 30.6       | 53.0 ± 47.7       | **1.1** ± 1.2     |
>
> &nbsp;
>
> 2. **"The number of test preferences seems slightly insufficient."**
>
> To enhance the empirical robustness of our experimental results in the traffic environment, we have increased the number of test preferences. We generated equidistant points on the 4-dimensional simplex using lattice-based discretization [3, 4]. In the initial draft, we set the granularity ($q$) to 3, resulting in $N_e = \frac{(q+m-1)!}{q!(m-1)!} = 20$ points. We have now increased the granularity to $q=4$, constructing $N_e = 35$ preferences. The updated results demonstrate that **our algorithm consistently outperforms the baseline methods in the traffic environment.**
>
> |                          | Base              | PCA               | AE                | NPCA              | **Ours**          |
> |--------------------------|-------------------|-------------------|-------------------|-------------------|-------------------|
> | **HV (× 10⁶¹, ↑)**       | 4.4 ± 6.8         | 0                 | 0.007 ± 0.018     | 19.4 ± 15.3       | **166.9** ± 48.1  |
> | **SP (× 10⁵, ↓)**        | 1842 ± 1290       | 3837 ± 2164       | 7834 ± 3323       | 34.2 ± 52.3       | **2.3** ± 1.0   |
>
>
> &nbsp;
>
> 3. **Regarding visualization in our high-dimensional reward space**
>
> To improve the presentation of our paper, we visualized the Pareto frontier obtained in the traffic environment for each algorithm using t-SNE. The corresponding figure has been added to the **(newly created) supplementary material**. NPCA overlaps with more points from the Base and AE methods than our method, demonstrating the insufficiency of NPCA in covering a larger Pareto frontier than the Base method. Additionally, most points generated by PCA are located on the opposite side of those produced by our method.

---

> ### Author Response · Authors · 2024-11-25
>
> 4. **"I am curious whether metrics like hypervolume and sparsity are sufficient."**
>
> To ensure that the current Pareto frontier $\mathcal{F}$ captures diverse return vectors in the original reward space, we recommend using an additional metric: the *Expected Utility Metric* (EUM) [5, 6]. EUM is defined as follows: $EUM(\mathcal{F}, f_s, \Omega_{K, \bar{N}_e})$
>
> = $\mathbb{E} [ \max_{r \in \mathcal{F}} f_s(\omega, r) ]$.
>
> Here, the expectation is calculated over $\omega \in \Omega_{K, \bar{N}_e} \subset \Delta^K$ that represents a set of $\bar{N}_e$ preferences in the original reward space, and $f_s$ is a scalarization function. We use EUM because it effectively evaluates the agent's performance across a wide range of preferences [6], aiming for a higher EUM to prevent the Pareto frontier from covering only a narrow region.
>
> To calculate EUM, we set $f_s(\omega, r) = \|\text{proj}_{\omega}[r]\|$, the projected length of the vector $r$ onto $\omega$. We also generated $\bar{N}_e$ equidistant points on the simplex $\Delta^K$ with granularity $q=5$, resulting in $\bar{N}_e = \frac{(5+5-1)!}{5!(5-1)!} = 126$ points for LunarLander-5D and $\bar{N}_e = \frac{(5+16-1)!}{5!(16-1)!} = 15,504$ points for the traffic enfironment.
>
> As demonstrated in the following tables, our method outperforms baseline approaches in terms of the EUM. It is important to note that during training, the 'Base' method uses equidistant points in the original reward space, which naturally leads to high EUM values, especially when the reward space has a high dimensionality. Nevertheless, our algorithm delivers superior performance compared to other reward dimension reduction methods, particularly in higher-dimensional environments like the traffic scenario. This advantage helps prevent the concentration of solutions in a narrow region within the original reward space.
>
> | LunarLander-5D           | Base              | PCA               | AE                | NPCA              | **Ours**          |
> |--------------------------|-------------------|-------------------|-------------------|-------------------|-------------------|
> | **EUM (↑)**              | -25.8 ± 24.3      | -20.2 ± 21.5      | -76.2 ± 48.6      | -28.4 ± 13.9      | **-11.5** ± 5.4   |
>
>
> | Traffic                  | Base              | PCA               | AE                | NPCA              | **Ours**          |
> |--------------------------|-------------------|-------------------|-------------------|-------------------|-------------------|
> | **EUM (× 10³, ↑)**       | -3.4 ± 2.9        | -35.1 ± 15.2      | -16.1 ± 8.8       | -4.4 ± 1.2        | **-2.0** ± 1.0    |
>
>
> &nbsp;
>
> References
>
> [1] Felten et al., "A toolkit for reliable benchmarking and research in multi-objective reinforcement learning", 2023.
>
> [2] Hung et al., "Q-Pensieve: Boosting Sample Efficiency of Multi-Objective RL Through Memory Sharing of Q-Snapshots", 2023.
>
> [3] Vijay K. Garg, "Introduction to Lattice Theory with Computer Science Applications", 2015.
>
> [4] "Simple-lattice designs", https://www.itl.nist.gov/div898/handbook/pri/section5/pri542.htm.
>
> [5] Zintgraf et al., "Quality Assessment of MORL Algorithms: A Utility-Based Approach", 2015.
>
> [6] Hayes et al., "A practical guide to multi-objective reinforcement learning and planning", 2022.

---

> > ### Comment · Reviewer_Pwfy · 2024-11-25
> >
> > I appreciate the authors' responses and additional experiments. However, I still have a few concerns:
> >
> > I agree with Reviewer iWoJ and I think the reason why previous MORL methods struggle in high-dimensional settings needs more clarification. Could it be due to the difficulty of covering all possible preferences during training?
> >
> > Second, I am confused about the additional visualization results. They do not seem to clearly reflect the dominance relationships between solutions. Additionally, the AE solutions appear more widely distributed—does this suggest better policy diversity? I also question whether a smaller overlap truly represents better diversity and seek clarification on what is meant by "located on the opposite side."
> >
> > While EUM performance is a helpful supplement, it may not fully reflect policy coverage since low-diversity solutions could still excel in EUM if they perform well on specific preferences. I acknowledge evaluating high-dimensional settings is challenging due to the lack of intuitive Pareto front visualizations so I recommend discussing this in future work section.

---

> ### Author Response · Authors · 2024-11-25
>
> We appreciate your prompt reply. Here are additional clarifications to your feedback.
>
> 1. **"Could it be due to the difficulty of covering all possible preferences during training?"**
>
> Yes. The challenge lies in effectively covering all possible preferences during training. In most previous MORL algorithms, agents sample random preferences in each episode to collect diverse behaviors. However, performing this sampling naively in high-dimensional spaces becomes computationally expensive because the coverage (or hypervolume) grows exponentially with the number of objectives [1]. To address this, our approach integrates dimension reduction techniques to down the search space while preserving the most relevant information. We will add this clarification explicitly in the final version of our paper.
>
> &nbsp;
>
> 2. **Clarification on visualization results**
>
> Our primary objective is to **cover a broad region of the true Pareto frontier, not merely to cover a wide region of a high-dimensional reward space itself**.
>
> Although AE solutions may appear widely distributed, this does not necessarily imply extensive coverage of the true Pareto frontier because the true Pareto frontier is a subset of the original space. Given that AE yields a low hypervolume, it is less likely to represent a wide range of the true Pareto frontier.
>
> Additionally, it is important to note that t-SNE is a nonlinear dimension reduction method [2] that preserves local structures rather than global relationships. Therefore, the dominance relationships in the original high-dimensional space may not be reflected as is in the visualization. However, t-SNE is useful because it maps high-dimensional data into a lower-dimensional space while maintaining local neighborhoods [2]. Thus, overlapping points in the t-SNE plot can be interpreted as indicating similar solutions.
>
> Regarding the overlap analysis, smaller overlaps with the Base method do not inherently indicate better diversity across the true Pareto frontier. They instead suggest a different local structure, for example, either in a way of our method (with high hypervolume) or a way of PCA (with very low hypervolume). This qualitative difference suggests that our method and PCA are distinct in their approach to exploring the solution space. We will also add this clarification in our final version.
>
> &nbsp;
>
> 3. **Regarding future work**
>
> We also acknowledge evaluating high-dimensional settings is challenging and there is still room for further research. We will include the discussion in our future work section.
>
> Thank you for your insightful feedback!
>
> &nbsp;
>
> References
>
> [1] Weijia Wang and Michèle Sebag, "Hypervolume indicator and dominance reward based multi-objective Monte-Carlo Tree Search", 2013.
>
> [2] Laurens van der Maaten and Geoffrey Hinton, "Visualizing Data using t-SNE", 2008.

---

> > ### Comment · Reviewer_Pwfy · 2024-11-26
> >
> > Thank you for the response, which addresses my concern. I would like to increase my score. I recommend that the author carefully discuss the limitations of existing work in high-dimensional settings and the necessity of dimensionality reduction techniques in future version.

---

> > > ### Author Response · Authors · 2024-11-26
> > >
> > > We sincerely thank the reviewer for raising the score to 8! In the final version, we will include all the discussions that you raised during the rebuttal.

---

### Author Response · Authors · 2024-11-27
**New revision uploaded**

Dear reviewers,

We thank the reviewers for valuable comments and constructive feedback, which have significantly enhanced the quality of our paper.

We have uploaded a revised version that addresses the points discussed during the rebuttal. The main changes are highlighted in blue throughout the text.

**Key updates include:**

1. **Strengthened Motivation:** We have expanded Section 3 (Related Work) to clarify the motivation behind our approach.

2. **Updated Main Results:** Section 5 (Experiments) now includes additional results to further support the empirical validity of our method.

3. **Expanded Discussions:** We have added comprehensive discussions on future work, visualization,  ablation study, and analyses related to the theorem and metrics in Appendices A, B, C, and D.

Thank you again for your insightful feedback!

---

### Author Response · Authors · 2024-11-28
**Final version uploaded**

Dear Reviewers,

We have uploaded the final version, which addresses the points discussed during the rebuttal.

Compared to the draft uploaded yesterday, we have refined the grammar and enhanced the overall presentation.

Thank you for your thorough and constructive feedback!

---

### Meta-Review · Area_Chair_FSw6 · 2024-12-10

**Metareview:**

The paper studies an important problem of reducing reward dimensions (i.e., number of objectives) in multi-objective RL. A neat solution is proposed which applies linear transformation to the reward. The authors theoretically established the conditions that guarantee the Pareto-optimality after this transformation. The proposed method is well tested on extremely challenging multi-objective decision making problems, showing significant performance improvements.

**Additional Comments On Reviewer Discussion:**

The reviewers raised some questions which are subsequently addressed by the authors. At the end of the rebuttal period, all reviewers unanimously agree that this is a good work and should be accepted.

---

### Decision · Program_Chairs · 2025-01-22

Accept (Poster)